# Airborne flux measurements of ammonia over the Southern Great Plains using chemical ionization mass spectrometry

Siegfried Schobesberger[1,2,3], Emma L. D'Ambro[4,*], Lejish Vettikkat[2], Ben H. Lee[1], Qiaoyun Peng[1], David M. Bell[5,6], John E. Shilling[5], Manish Shrivastava[5], Mikhail Pekour[5], Jerome Fast[5], Joel A. Thornton[1]

[1]Department of Atmospheric Sciences, University of Washington, Seattle, WA, USA.
[2]Department of Applied Physics, University of Eastern Finland, Kuopio, Finland.
[3]Department of Physics, University of Helsinki, Helsinki, Finland.
[4]Department of Chemistry, University of Washington, Seattle, WA, USA.
[5]Pacific Northwest National Laboratory, Richland, WA, USA.
[6]Laboratory of Atmospheric Chemistry, Paul Scherrer Institute, Villigen, Switzerland
[*]*Now at*: Office of Research and Development, U.S. Environmental Protection Agency, Research Triangle Park, NC, USA.

*Correspondence to*: Siegfried Schobesberger (siegfried.schobesberger@uef.fi)

**Abstract.** Ammonia ($NH_3$) is an abundant trace gas in the atmosphere and an important player in atmospheric chemistry, aerosol formation and the atmosphere-surface exchange of nitrogen. The accurate determination of $NH_3$ emission rates remains a challenge, partly due to the propensity of $NH_3$ to interact with instrument surfaces leading to high detection limits and slow response times. In this paper, we present a new method for quantifying ambient $NH_3$, using chemical ionization mass spectrometry (CIMS) with deuterated benzene cations as reagents. The setup aimed at limiting sample-surface interactions and achieved a 1-$\sigma$ precision of 10-20 pptv and an immediate $1/e$ response rate < 0.4 s, which compares favorably to the existing state of the art. The sensitivity exhibited an inverse humidity dependence, in particular in relatively dry conditions. Background of up to 10% of the total signal required consideration as well, as it responded on the order of a few minutes. To showcase the method's capabilities, we quantified $NH_3$ mixing ratios from measurements obtained during deployment on a Gulfstream I aircraft during the HI-SCALE (Holistic Interactions of Shallow Clouds, Aerosols and Land Ecosystems) field campaign in rural Oklahoma during May 2016. Typical mixing ratios were 1-10 parts per billion by volume (ppbv) for the boundary layer and 0.1-1 ppbv in the lower free troposphere. Sharp plumes of up to tens of ppbv of $NH_3$ were encountered as well. We identified two of their sources as a large fertilizer plant and a cattle farm, and our mixing ratio measurements yielded upper bounds of $350 \pm 50$ and 0.6 kg $NH_3$ $h^{-1}$ for their respective momentary source rates. The fast response of the CIMS also allowed us to derive vertical $NH_3$ fluxes within the turbulent boundary layer via eddy covariance, for which we chiefly used the continuous wavelet transform technique. As expected for a region dominated by agriculture, we observed predominantly upward fluxes, implying net $NH_3$ emissions from surface. The corresponding analysis focused on the most suitable flight, which contained two straight-and-level legs at ~300 m above ground. We derived $NH_3$ fluxes between 1 and 11 mol $km^{-2}$ $h^{-1}$ for these legs, at an effective spatial resolution of 1-2 km. The analysis demonstrated how flux measurements benefit from suitably arranged flight tracks with sufficiently long straight-and-level

legs, and explores the detrimental effect of measurement discontinuities. Following flux footprint estimations, comparison to the NH$_3$ area emissions inventory provided by the US Environmental Protection Agency indicated overall agreement, but also the absence of some sources, for instance the identified cattle farm. Our study concludes that high-precision CIMS measurements are a powerful tool for in-situ measurements of ambient NH$_3$ mixing ratios, and even allow for the airborne mapping of the air-surface exchange of NH$_3$.

## 1 Introduction

Ammonia (NH$_3$) is the most abundant alkaline gas in the atmosphere, with mixing ratios ranging from < 10 parts per trillion by volume (pptv) in very remote regions to tens of parts per billion by volume (ppbv) in areas with high anthropogenic emissions (e.g., Norman and Leck, 2005; Shephard et al., 2020; Wu et al., 2021; Zhu et al., 2022). Consequently, it plays an important role in atmospheric and environmental chemistry as well as atmosphere-ecosystem relations: from the formation of inorganic and organic aerosol, to soil acidification and nutrient cycles.

NH$_3$ is a key player in the atmosphere-ecosystem exchange and biogeochemical cycling of nitrogen (N). Agricultural soils are typically deficient in N as a nutrient for plant growth, leading to the copious use of NH$_3$ as fertilizer and related N fertilizers such as urea. Volatilization of NH$_3$, in particular from NH$_4$-forming fertilizers, is in turn a major N loss mechanism for agricultural soils (Ma et al., 2021) while constituting a major source of atmospheric NH$_3$. Agricultural activities are indeed the dominant source of atmospheric NH$_3$. Of particular importance is also livestock farming (in particular pig, cattle, poultry) and manure processing. On the other hand, ecosystem exposure to and uptake of NH$_3$ (e.g., via dry deposition) are associated with numerous adverse environmental effects (e.g., via conversion to nitrous oxide, a greenhouse gas, or nitrate, which may leach into water) and biological effects, in particular on native vegetation (Krupa, 2003). Critical NH$_3$ thresholds (Cape et al., 2009) are exceeded across Europe (Tang et al., 2021), and contributes to critical reactive N load exceedances in North America (Walker et al., 2019).

In the atmosphere, NH$_3$ contributes to aerosol particle formation, in particular by associating with nitric acid (HNO$_3$) to form ammonium nitrate (NH$_4$NO$_3$), which can dominate the inorganic pollution load (Tang et al., 2021; Bressi et al., 2021). NH$_3$ emissions thereby contribute substantially to fine-particle pollution and may make agriculture the leading air pollution source to contribute to premature mortality in Europe and parts of North America and Asia (Lelieveld et al., 2015). Solid NH$_4$NO$_3$ particles have also been detected in the upper troposphere, where they may play an important role as ice nuclei (Höpfner et al., 2019). Furthermore, NH$_3$ is implicated in the first steps of new-particle formation, in particular by association with sulfuric acid (Schobesberger et al., 2013), and expected to play an important role in organic-poor environments such as the upper troposphere (Dunne et al., 2016). NH$_3$ may limit new-particle formation also in Antarctica (Jokinen et al., 2018), and it is implicated in intensive local cluster formation events that were observed over agricultural fields (Olin et al., 2022). Typical atmospheric aerosol also contains a major, chemically complex organic component, which

likely also facilitates reactive uptake of $NH_3$, forming salts or N-containing organics (e.g., Liu et al., 2015; Bell et al., 2017; Wu et al., 2021).

Despite its importance, there are high uncertainties in attributing atmospheric $NH_3$ to specific sources, and current emissions inventories for $NH_3$ are suspected to have large uncertainties (Vonk et al., 2016; Grönroos et al., 2017; EEA, 2019). A major reason is also practical difficulties in establishing $NH_3$ emissions by concentration measurements, as bottom-up approaches

may not capture all sources, and top-down approaches may not capture the total emissions, especially for outdoor farming activities or naturally ventilated buildings (e.g., Calvet et al., 2013; Oliveira et al., 2021). Also, some sources of $NH_3$ may not be understood well enough. E.g. for urban environments, catalytic converters in vehicles have been recognized as a source of $NH_3$ that is likely grossly underrepresented in current emissions inventories (Sun et al., 2017; Farren et al., 2020). Correspondingly, observations tend to indicate that $NH_3$ emissions are substantially underestimated; e.g., airborne

measurements in Utah, where $NH_4NO_3$ plays a major role in pollution (Franchin et al., 2018; Moravek et al., 2019a). Globally, satellite data have recently revealed hundreds of small-area (< 50 km) or point sources to be mostly underrepresented in emissions inventories by even more than an order of magnitude (Van Damme et al., 2018). Strong day-to-day variability was found as well (Fortems-Cheiney et al., 2016). Overall, however, observational data on $NH_3$ concentrations are scarce, limiting evaluations of model simulations, such as models of $NH_3$ emissions and aerosol

formation. Vertically resolved observational data, ideally using airborne in-situ measurements, are sparser still.

The accurate quantification of $NH_3$ emissions and concentrations is challenging, due to the wide range of ambient mixing ratios and its infamous propensity to interact with sampling and instrument surfaces, causing losses and slow response times. A wide variety of techniques have to date been used to quantify $NH_3$ mixing ratios. Typical precisions and detection limits are tens of pptv at best, and time responses often on the scales of minutes (von Bobrutzki et al., 2010). Such performance

limitations can lead to substantial errors when low or fast-changing concentrations are to be captured accurately: e.g., plumes, mobile deployments, remote locations, free troposphere; or for applying the eddy covariance method (EC) to measure vertical exchange (Moravek et al., 2019b) from which emission rates can be derived. Some optical and mass spectrometry techniques have pushed these boundaries and offer fast response, while also allowing deployment on aircraft or for EC. Most airborne in-situ measurements of $NH_3$ have been during a number of aircraft campaigns, mostly in the US, that

have deployed chemical ionization mass spectrometers (CIMS; Nowak et al., 2007; Nowak et al., 2010; Nowak et al., 2012) or infrared laser spectrometry techniques. Examples for the latter are off-axis integrated cavity output spectrometry (off-axis ICOS; Leen et al., 2013), and, predominating more recently, tunable infrared laser differential absorption spectrometry (TILDAS; Moravek et al., 2019a; Pollack et al., 2019). Critical for fast instrument response times are design elements that reduce interactions between sample and surfaces of the sampling setup and the instrument, e.g. a shortened and straightened

sampling line, high sampling flow and shortened reaction chamber in airborne CIMS deployments (Nowak et al., 2010), or a heated high-flow sampling line with active continuous passivation in airborne TILDAS deployments (Pollack et al., 2019).

For measuring surface-atmosphere exchange rates (e.g., emission or dry deposition), or vertical fluxes more generally, EC has been established as one of the most direct techniques. EC relies on measuring both the fluctuations of the vertical wind

component caused by the turbulence in the atmospheric boundary layer, and the simultaneous fluctuations of a scalar
magnitude, such as temperature or a vapor's mixing ratio. If the surface, or possibly the air below the measurement height, is
a net source or sink for the scalar, these fluctuations will correlate positively or negatively, and their covariance is a direct
measure of the vertical flux of the scalar at the measurement height. When the surface is the only net source or sink, that
vertical flux is assumed constant with height within the surface layer, i.e., up to ~100 m, and typically decreases linearly with
height (referred to as vertical flux divergence) within the core boundary layer above (e.g., Lenschow et al., 1980). Under
these conditions and assumptions, airborne flux measurements can directly infer rates of net emission and dry deposition.
Consequently, airborne EC has been applied for more than 30 years (e.g., Lenschow et al., 1980; Lenschow et al., 1981;
Desjardins et al., 1982; Ritter et al., 1990; Ritter et al., 1992; Ritter et al., 1994; Dabberdt et al., 1993). Studies over the last
10-15 years have developed continuous wavelet transform (CWT) analysis to calculate spatially resolved fluxes from
airborne measurements (e.g., Mauder et al., 2007; Karl et al., 2009; Metzger et al., 2013; Karl et al., 2013; Misztal et al.,
2014; Yuan et al., 2015; Wolfe et al., 2015; Vaughan et al., 2017; Sayres et al., 2017; Desjardins et al., 2018; Wolfe et al.,
2018; Hannun et al., 2020), including dedicated aircraft campaigns (e.g., BOREAS, CABERNET, CARAFE, OPFUE) and
platforms (e.g., FOCAL). Spatial resolutions of a few km are typically achieved with good accuracy.

A key challenge for successful EC flux measurements is the requirements for fast time response and high precision, in order
to capture the full range of turbulence timescales; similar to the desired performance of airborne measurements more
generally. A sampling rate of 10 Hz is typically desired for surface-layer eddies. One way of achieving fast instrument
response is the use of open-path sensors, which practically eliminate interactions between sample and instrument surfaces
altogether. E.g., an open-path sensor that measures $NH_3$ via absorption of a quantum cascade infrared laser offers a precision
of 150 pptv at >1 Hz (1-$\sigma$) and has been successfully deployed to measure EC fluxes (Miller et al., 2014; Sun et al., 2015),
also in low-$NH_3$ environments (Pan et al., 2021). Airborne deployments have so far favored closed-path systems, as
introduced above. They tend to achieve similar or better 1-Hz precisions, in particular when extra attention is paid to
reducing sample-surface interactions in the sampling setup. E.g., Pollack et al. (2019) reported an Allan deviation (1-$\sigma$) of 60
pptv for their airborne TILDAS with optimized sampling. Airborne CIMS deployments for measuring $NH_3$ have achieved
comparable precision, e.g., 80 pptv (1-Hz, 1-$\sigma$) was reported for the acetone-CIMS deployments on the WP-3D aircraft of
the National Oceanic and Atmospheric Administration (NOAA) (Nowak et al., 2012).

Over the last decade, developments in the application of mass spectrometry for ambient measurements, e.g. using CIMS
techniques, have greatly improved our capabilities in identifying and quantifying atmospheric trace gases. State-of-the-art
time-of-flight (TOF) mass spectrometers typically feature a versatile atmospheric pressure interface that efficiently transmits
ions from a high-pressure (up to atmospheric) or low-vacuum (> 1 mbar) ion source to the high-vacuum (< $10^{-5}$ mbar) TOF
region that facilitates identification and detection (Junninen et al., 2010; Jokinen et al., 2012). The high-pressure ion source
may be a simple chamber, typically held at > 100 mbar, in which reagent ions are admixed to the analyte sample to
chemically ionize target compounds (ion-molecule reaction region, IMR). That setup readily allows for using a variety of
different reagent ions, common examples being acetate, iodide and bromide anions (e.g., Bertram et al., 2011; Lee et al.,

2014; Sanchez et al., 2016), and water, benzene and toluene cations (e.g., Aljawhary et al., 2013; Kim et al., 2016; Alton and Browne, 2020). For efficiently ionized compounds, these TOF-CIMS devices achieve limits of detection down to < 1 pptv and 1-Hz precisions (1-$\sigma$) < 10 pptv (e.g., Bertram et al., 2011). Consequently, they have also been used to measure surface-layer EC fluxes of a variety of compounds (e.g., Nguyen et al., 2015; Schobesberger et al., 2016; Fulgham et al., 2019). A potential key advantage of TOF mass spectrometers is their high acquisition rate of full mass spectra. They can be readily recorded at 10 Hz or more, until sensitivity or data storage become practical limitations. By routinely counting a wide range of ions simultaneously, the mixing ratios or EC fluxes of multiple compounds can in principle be quantified from the same datasets. Optical devices, on the other hand, may be limited to quantifying single compounds only, or a handful at best. Several chemical ionization schemes have been used for detecting $NH_3$ in the past, including acetone-CIMS (Nowak et al., 2012), ethanol-CIMS (You et al., 2014) and water-cluster-CIMS (Zheng et al., 2015; Pfeifer et al., 2020).

In this paper, we present a new TOF-CIMS method for quantifying ambient $NH_3$ mixing ratios, using as reagents deuterated benzene cations ($C_6D_6^+$) and their clusters (e.g., dimer cations ($C_6D_6$)$_2^+$). The use of benzene-CIMS was motivated by its capability of detecting terpenes, important biogenic volatile organic compounds (Leibrock and Huey, 2000; Kim et al., 2016; Lavi et al., 2018), and deuterated benzene was chosen to aid in differentiating ion compositions containing reagents, e.g. via adduct formation. We deployed the instrument on the Gulfstream I (G-1) aircraft of the U.S. Department of Energy's Atmospheric Radiation Measurement (ARM) Aerial Facility during the Holistic Interactions of Shallow Clouds, Aerosols and Land Ecosystems (HI-SCALE) field campaign in Oklahoma in 2016 (Fast et al., 2019). This paper hence demonstrates the suitability of our benzene-CIMS setup for airborne measurements. Moreover, the datasets proved suitable for calculating vertical fluxes via EC, making use of the 3-D wind data obtained by turbulence probes also carried by the G-1. To our knowledge, we are thereby reporting the first use of TOF-CIMS for measuring EC fluxes of $NH_3$. We explore that capability of the instrument, including the use of the CWT method, as well as the capability of the airborne eddy flux data to infer area emission rates of $NH_3$ attributed to agriculture in rural Oklahoma.

## 2 Methods

### 2.1 Measurement technique

We added benzene-CI capability to an existing CIMS setup designed to use iodide anions as reagents. Iodide-CI remained the instrument's primary mode of operation. By switching instrument voltage polarities, iodide-CI and benzene-CI could be used alternatingly. At its core, the CIMS consists of a high-resolution TOF mass analyzer and a sequentially pumped atmospheric-pressure interface (APi) to guide and focus ions from a pressure-controlled ion-molecule reaction region (IMR) towards the high vacuum in the TOF region (Aerodyne Research Inc., USA, and Tofwerk AG, Switzerland). Detailed descriptions thereof are found in existing literature (e.g., Junninen et al., 2010; Lee et al., 2014). The specific CIMS at hand had been configured to fit into an aircraft rack for deployment on the National Science Foundation's C-130 aircraft and featured some modifications to allow for efficient sampling and quantifiable measurements during airborne deployments.

These modifications are described in detail in Lee et al. (2018). They include a computer-controlled variable orifice to maintain a constant sample mass flow rate into the IMR, a port at the orifice to inject clean gas for determining instrument backgrounds. A high-flow inlet with sub-sampling from the centerline into a shortened IMR reduced vapor-wall interactions. For deployment on the G-1, the configuration differed slightly: (i) a machined PTFE cup was press-fit into the IMR to further reduce vapor-wall interactions; (ii) the CIMS was isolated from vibrations and (some) shocks by mounting it on wire-rope isolator inside its rack, whereas most accessories (most electronics, pumps, flow controllers, etc.) as well as the rack itself were not; (iii) the inlet tube was fastened to the aircraft fuselage but connected to the instrument via an Ultra-Torr fitting that allowed for motion relative to the instrument if forced; (iv) the inlet tip was simply cut straight and faced perpendicular to the airflow. The aircraft cabin was not pressurized.

Setup, flows and pressures of the CIMS inlet system are schematically shown in Fig. 1. Ambient air was sampled at 22 L min$^{-1}$ through the cabin wall via a straight, 40-cm long, 3/4-inch (1.9 cm) outer diameter (OD), 1.6 cm inner diameter (ID) Teflon tube. Most of that inlet flow was provided by a dedicated sample pump (Vaccubrand MD1) and discarded; 2000 sccm entered the IMR through the variable orifice. The pressure in the IMR was maintained at 100 mbar by a servo-controlled valve throttling a dedicated scroll pump (Agilent IDP-3). The measured mass flow of the pump exhaust was used to control the variable orifice to maintain the 2000-sccm sample flow into the IMR. An additional flow of 1500 sccm nitrogen (ultra-high purity $N_2$, Scott-Marrin or Airgas, UHP grade 5.0) carried methyl iodide ($CH_3I$, from a permeation device) and deuterated benzene ($C_6D_6$) through an ionizer ($^{210}$Po, 10 mCi, NRD) into the IMR. Nominal benzene mixing ratios in the IMR of above 100 parts per million by volume (ppm) were desired to obtain high, stable sensitivities in benzene-CI mode (Lavi et al., 2018), which was achieved by diverting 10 sccm out of the 1500-sccm ionizer flow over the headspace of about a mL of $C_6D_6$ (Cambridge Isotope Laboratories, Benzene-D6, D, 99.5%) in a glass test tube. The 10:1490 flow ratio would momentarily be increased to 100:1400 during instrument start-up to speed up conditioning the lines and obtain stable reagent ion counts. A pair of manual shut-off valves isolated the benzene when not in use. A needle valve in its fully open position served as a critical orifice (1 L min$^{-1}$ at 1 atm) upstream of the $CH_3I$ permeation device, keeping the headspace of the benzene at ~1.5 atm to slow down its vaporization (and indeed prevent its boiling under otherwise 100 mbar). The benzene was not temperature controlled. With the G-1 cabin temperature ranging from 15 to 30 °C, the calculated $C_6D_6$ mixing ratios in the IMR ranged from 160 to 360 ppm, nominally, i.e., assuming the headspace was saturated (thermodynamics data from Zhao et al., 2008). The benzene consumption rate appeared to suggest somewhat lower actual mixing ratios, but it was not systematically monitored. IMR background signals ("zeros") were determined every 42 s by overflowing the variable orifice with 2200 sccm of $N_2$ for 6 s (Fig. 1). While the IMR was actively humidified in iodide-CI mode to reduce variations of sensitivities as a function of ambient humidity (Lee et al., 2014; Lee et al., 2018), that humidification was turned off for operations in benzene-CI mode, starting from research flight 5 (RF5). The humidification caused spurious spikes in signals of interest (e.g., $C_{10}H_{16}^+$) that we could not otherwise dispose of in the field.

Typical reagent ion count rates were 2-3 × 10$^6$ cps of $C_6D_6^+$ and 2-4 × 10$^5$ cps of $(C_6D_6)_2^+$. Also observed during in-flight operation were $CH_3I.C_6D_6^+$, $H_2O.H_3O^+$, $(H_2O)_2.H_3O^+$ and $(H_2O)_3.H_3O^+$, typically below 1, 2, 3 and 0.5 × 10$^4$ cps,

respectively. NH$_3$ was quantified from the normalized count rate of NH$_3$.C$_6$D$_6^+$ adduct ions. Normalization was to 10$^6$ cps of C$_6$D$_6^+$, i.e., measured counts per second (cps) of NH$_3$.C$_6$D$_6^+$ would typically be divided by a factor of 2-3 to obtain normalized counts per second (ncps). Only the C$_6$D$_6^+$ signal was used in normalization, as NH$_3$.C$_6$D$_6^+$ responded more directly to changes in C$_6$D$_6^+$ rather than (C$_6$D$_6$)$_2^+$ or the sum of both.

NH$_4^+$ was detected as well, but with ~2 orders of magnitude lower counts. As its signal clearly covaried with the protonated water cluster signals, NH$_4^+$ was likely formed by proton transfer from water and of no further interest here.

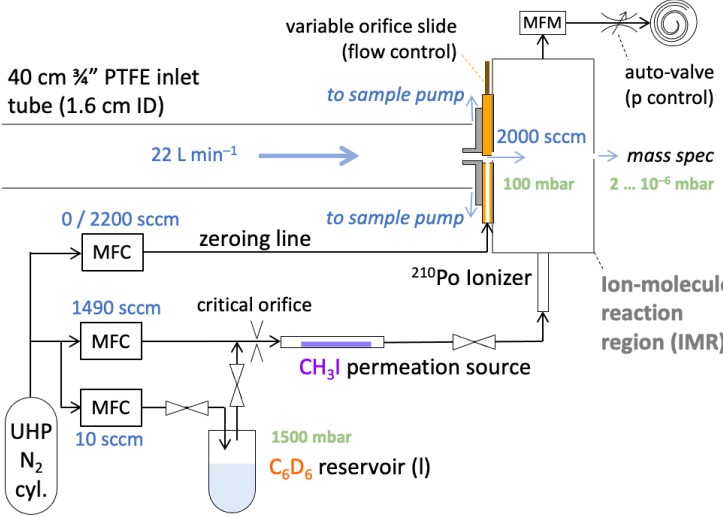

**Figure 1: Schematic of the sampling setup and the flows into IMR of the CIMS. The flows into the IMR are 1500 sccm of ultra-high purity (UHP) N$_2$, set by mass flow controllers, and 2000 sccm of sample, maintained by the variable orifice and controlled by mass flow controllers (MFC) and mass flow meter (MFM) measurements (details in text). IMR pressure is maintained at 100 mbar by the auto-valve in the pumping line. A fraction of the UHP N$_2$ passes over the headspace of a reservoir of liquid C$_6$D$_6$ under ~1500 mbar due to a critical orifice; all N$_2$ then passes over a CH$_3$I permeation device. Just prior to entering the IMR, a $^{210}$Po ionizer provides the primary reagent ions: I$^-$, C$_6$D$_6^+$ and (C$_6$D$_6$)$_2^+$.**

## 2.2 Field campaign

The results reported in this paper are taken from measurements during the CIMS' deployment on the ARM Aerial Facility's G-1 aircraft for the HI-SCALE field campaign. An overview of the HI-SCALE campaign is provided in Fast et al. (2019), including measurement approach and descriptions of the instrumentation deployed on the G-1 besides the CIMS. We focus here specifically on the first of the two intensive operating periods (IOP1), during which research flights were performed between 24 April and 21 May. The aircraft was based out of Bartlesville Municipal Airport (airport code KBVO), 60 km north of Tulsa, Oklahoma, USA. The research flights, however, were concentrated around the ARM Southern Great Plains Central Facility ground site (SGP), located 130 km to the east of KBVO, or 120 km north of Oklahoma City, at an elevation of 314 m above mean sea level (AMSL). The flight profiles consisted mostly of patterns of vertically stacked straight-and-level legs as well as ascending or descending straight transects or spirals. Vertically, the flights focused on the region around shallow clouds, from the middle of the boundary layer to the lower free troposphere, as the campaign goal was to study

surface-aerosol-cloud interactions. Ground speeds were typically between 80 and 110 m s$^{-1}$. Occasionally, straight-and-level legs were flown as low as ~300 m above ground. Those legs were our focus for eddy covariance analysis, especially when

flown broadly perpendicular to the wind. Figure S1 provides a geographical overview of the campaign environment and of the flight portions when operating in benzene-CI mode.

The CIMS was installed on the port side, in the front section of the main cabin, and sampled straight through the port cabin wall. The CIMS was turned on several hours before each research flight to allow conditions to stabilize before departure and to perform calibrations. Power typically continued to be available for several hours after landing, allowing for some more

checks and maintenance, but the CIMS needed to be shut down at the conclusion of each workday, as the aircraft remained unattended and unpowered overnight. During research flights, the CIMS recorded full mass spectra at a frequency of 2 Hz. A data acquisition rate of 2 Hz, rather than e.g. 10 Hz, was a compromise that reduced requirements for data storage and computing times during data processing, while potentially induced errors in obtained fluxes were deemed acceptable (see Sect. 3.8 for details).

For the eddy covariance analysis in this study, airborne meteorological data was provided by the Aircraft-Integrated Meteorological Measurement System (AIMMS-20, Aventech Research Inc., Canada) that was mounted on the side of the nose of the aircraft. The AIMMS-20 measured temperature, relative humidity (RH) and static pressure at nominally 20 Hz, and calculated 20-Hz 3-D ambient wind based on measurements of the differential pressures from a 5-port hemispheric gust probe, aircraft position, velocity and attitude (Beswick et al., 2008). For ambient temperature and RH, however, we used

static air and dew point temperature data obtained by the Rosemount 102E Pt100 sensor and General Eastern 1011-B chilled-mirror hygrometer, respectively. These data were nominally only 1-Hz but exhibited a better real time response than the corresponding AIMMS-20 data.

All times in this study are in Coordinated Universal Time (UTC), which was five hours ahead of the locally observed Central Daylight Time (CDT). Terrain elevation data was retrieved from a digital elevation model (Yamazaki et al., 2017).

**2.3 Calibration setups**

**(1)** Initial experiments for determining the CIMS sensitivity to $NH_3$ (along with isoprene, α-pinene and dimethyl sulfide) were carried out in the ARM Aerial Facility's hangar in Pasco, WA, during summer 2016. The instrument was in the same place and configuration (i.e., in the aircraft) as during the field campaign. A wafer-type permeation device was used as the source of $NH_3$ (Type 40F3, VICI Metronics Inc., Poulsbo, WA, USA; 41 ± 10 ng/min at 30 °C). We continuously kept the

source 40 °C and gravimetrically measured a permeation rate of 46 ± 4 ng/min. For the experiments in Pasco, a flow of 1 slpm $N_2$ over the $NH_3$ source was diluted by a larger flow of $N_2$ (up to 10 slpm) and the total was directed at the CIMS inlet outside the aircraft. The sample pump was turned off, which reduced the total inlet flow to 2 slpm, and the inlet was thus overblown by dry $N_2$ containing $NH_3$ as adjusted by the dilution ratio.

**(2)** Follow-up experiments took place in the lab (Sept.-Oct. 2016) and used lab air and dry or humidified $N_2$. First, a $N_2$ flow

over the permeation source of between 0.2 and 1 slpm was simply directed at the IMR orifice via a short 19-mm OD Teflon

line, thereby effectively diluting with lab air (~40% RH) at ratios of 1.8:0.2 to 1:1, which varied $NH_3$ mixing ratios while resulting in an RH range from 20% to 36% for that experiment. In the next set of lab experiments, the humidity dependence was more systematically explored by overflowing the inlet with 2.2 slpm of humidified $N_2$ passing over the permeation source. Measured RH values of <1%, 15.5% and 52.5% (Vaisala HM34) were obtained by optionally bubbling part of the $N_2$ through water.

**(3)** Another set of experiments was conducted during spring 2022, to further explore the dependence of sensitivity to $NH_3$ on humidity, as well as the possible influences of carrier gas ($N_2$ vs. air) and ionizer flow composition: $C_6D_6$ vs. $C_6H_6$ (anhydrous, 99.8%, Sigma-Aldrich), and $CH_3I$ vs. no $CH_3I$. For those experiments, a different CIMS was used. That instrument was largely identical to the CIMS used in 2016. The most important difference was the use of the commercially available stainless-steel IMR (Bertram et al., 2011; Aerodyne Research) with a simple 10-mm OD stainless-steel inlet port. Inlet and ionizer flows were 2 and 1 slpm, respectively; there was no make-up flow. A newly purchased $NH_3$ permeation source was press-fit into a hole on the side of a short 10-mm polyoxymethylene tube, which was attached to the CIMS inlet. The source was not heated but subject to the temperature in the lab, which was air-conditioned to 23 °C. Its gravimetric permeation rate was $11 \pm 3$ ng/min. Various $NH_3$ mixing ratios were delivered to the instrument by overflowing with $N_2$ or purified air (either optionally humidified) and venting either before or after the source.

# 3 Results & Discussion

## 3.1 Calibration

The initial calibration experiments, using dry $N_2$ (1), resulted in a measured sensitivity of only ~0.5 npcs/pptv, which was at least an order of magnitude lower than roughly expected from the campaign data. However, that result was consistent with follow-up calibrations in dry $N_2$ or dry air (2+3). For the follow-up experiments using lab air for dilution, the sampled RH varied between around 20% and 36% between individual measurements. The results were consistent with a sensitivity of 4.2 ncps/pptv, as obtained by a line fit (Fig. S2a), i.e., no humidity dependence was apparent. The experiments using humidified $N_2$ (RH > 15%) yielded similar sensitivities, but again a much lower sensitivity (0.54 ncps/pptv) for dry $N_2$. The resulting picture was that of a relatively low humidity dependence for RH > ~20%, but a sharp drop in sensitivity under dry conditions (Fig. 2, orange).

Later lab experiments using another CIMS device (3) yielded a similar humidity dependence (Fig. 2, blue). Figure 2 shows measured sensitivities against absolute humidity in the IMR, along with empirical exponential fits. Plotting against *sample* humidity (absolute or relative) would reduce the gap between the curves, as the earlier experiments (orange) used a 50% larger ionizer flow, implying a smaller absolute humidity in the IMR compared to the later experiments (blue) for any given sample humidity. As the CIMS devices (blue vs. orange) differed in particular in their IMR geometries, we leave it up to future studies to ascertain if sensitivity to $NH_3$ relates to the humidity of only the sample (absolute or relative) or the humidity in the IMR.

In the later lab experiments (3), we also examined the sensitivity to NH$_3$ when using regular, non-deuterated benzene (C$_6$H$_6$) instead of C$_6$D$_6$, when the CH$_3$I permeation source was removed, and both. All variations led to slightly lower sensitivities,

but not significantly so, in particular given possible variability in the NH$_3$ permeation rates due to the source not being temperature-controlled.

The supplement contains further details on our calibration experiments, including Fig. S2.

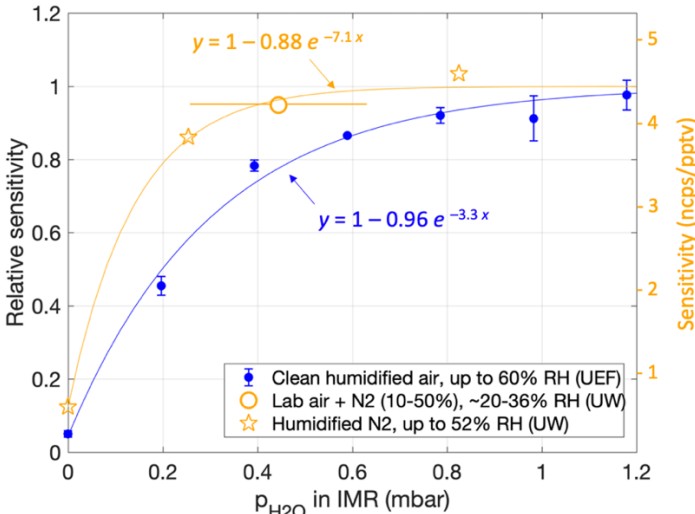

**Figure 2: Humidity dependence of the sensitivity of NH$_3$.C$_6$D$_6^+$ to the NH$_3$ mixing ratio, shown as a function of water partial**
**pressure in the IMR as calculated from the humidity in the sample flow. Ion counts were normalized to the total reagent ion counts ([C$_6$D$_6^+$]). Orange markers are from calibration experiments at the University of Washington (UW) lab immediately after the HI-SCALE campaign. Bars for the lab air result indicate a sample RH range from 16% to 40% (i.e., including 4% uncertainty); sample RH for results in N$_2$ (stars) were 0%, 16% and 52%. The right-hand scale also reports the corresponding sensitivities obtained from a temperature-controlled NH$_3$ permeation source. Blue markers are from experiments using a largely**
**corresponding setup but with a different CIMS device at the University of Eastern Finland (UEF). Corresponding sample RH values are from 0% to 60% in steps of 10%, i.e., divergent from the UW results due to a smaller (dry) ionizer flow into the IMR. Lines are weighted fits of the form $y = 1 - ae^{-bx}$.**

## 3.2 Response times and precision

We investigated the CIMS response to changes in NH$_3$ mixing ratios by analyzing 'zero' measurements, i.e., transitions from
ambient (NH$_3$ levels > 1 ppbv) to instrumental backgrounds. Figure 3 presents three such occasions at 1-s resolution. The dark blue crosses are normalized count rates of NH$_3$.C$_6$D$_6^+$ before and during overflowing the tip of the inlet line with >26 lpm of dry N$_2$. The signal, first corresponding to an ambient NH$_3$ mixing ratio of 2.9 ppbv, dropped rapidly, by 90% within 1 s, at an immediate $1/e$ response rate of 0.25 s. Part of that drop, however, needed to be due to the decreased sensitivity with a dry IMR (cf. Fig. 2). The blue shade indicates the theoretical 'worst case' of an immediate sensitivity drop. It implies a
somewhat higher than apparent background and a fast drop possibly by only 80%, though the worst-case immediate $1/e$ response rate would still be < 0.4 s. Magenta crosses (Fig. 3b) show the response to an in-flight zero at the IMR orifice, which dropped yet more rapidly and steeply, probably because it was subject to mixing and equilibration processes only in

the IMR but not in the 40-cm inlet line. The steeper drop thus suggests that the sampling inlet played a role in the instrument's response besides the IMR, at least for time scales longer than 1 s. Most likely, $NH_3$ was partitioning back from

the walls of the sampling line, creating an elevated background. The level of that background would then be related to previously sampled $NH_3$ mixing ratios. That background level is responding more slowly. For three long 'zeros' at the sampling inlet tip, that slower decay followed a time constant of $4 \pm 2$ min.

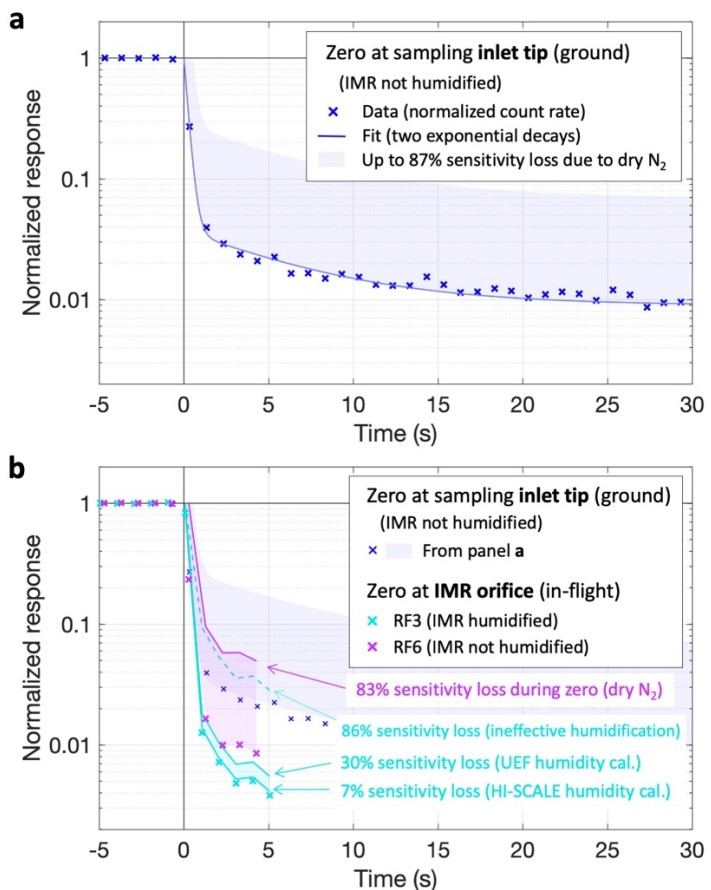

**Figure 3: Responses to instrument "zeros" by overflowing dry $N_2$. Blue: A zero at the sampling inlet tip performed on the ground**
**(pre-flight, RF16, 19 May, ambient [NH₃] = 2.9 ppb); crosses are normalized count rates; the line (panel a only) presents a fit of these data by the sum of two exponential decays with time constants of 0.24 and 6.4 s; the shaded area is for multiplying the fit by up to a factor of 7.7 (except for unity as the upper bound) corresponding to the loss of sensitivity observed in dry $N_2$ vs. ambient air (Figs. S2c and 2). Panel b compares these results to examples for in-flight zeros at the IMR orifice (magenta and cyan), randomly chosen among momentarily stable [NH₃] signals during RF6 (17:24 UTC, 3 May, ambient [NH₃] = 3.9 ppb) and RF3**
**(21:44 UTC, 28 April, ambient [NH₃] = 2.5 ppb). Crosses are again normalized count rates; lines and shades indicate underestimations of zero signals due to drying by the $N_2$ overflowing the IMR orifice following Fig. 2. Unlike most flights, the IMR was continuously humidified via a separate line during RF3 (cyan); the dashed line indicates the underestimation in case the IMR humidification was entirely ineffective in maintaining sensitivity to $NH_3$.**

Figure 3b also shows an in-flight zero from an earlier flight during which the IMR was actively humidified (cyan).
Interestingly, the response here was even better, even though the concurrent drop in sensitivity was expected to be much

smaller (cf. Fig. 2), We attributed this observation to flight-to-flight variability, as two separate tests of turning the IMR humidification on during inlet-tip zeros (dry $N_2$, on the ground) did multiply the $NH_3.C_6D_6^+$ background count rate by factors of 9 and 16. That observation is consistent with active IMR humidification aiding in maintaining sensitivity to $NH_3$, plus some contaminant $NH_3$.

We examined the instrument's precision based on the pure background signal during four of the longer zero measurements by overflowing the inlet tip with dry $N_2$ on the ground (e.g., Fig. 3, dark blue). We obtained 1-Hz precisions of 5 to 11 ncps (1-$\sigma$), or in terms of mixing ratios, 10 to 20 pptv. A customary definition of the limit of detection (LOD) is three times the 1-$\sigma$ precision, yielding a 1-Hz LOD in the range of 30 to 60 pptv. This is on par with or better than high-performance $NH_3$ detectors previously deployed on aircraft (see Introduction).

**3.3 Quantification**

The first step for obtaining $NH_3$ mixing ratios from the CIMS measurements was normalization of $NH_3.C_6D_6^+$ count rates to $C_6D_6^+$ primary ion count rates. The high-frequency (1-Hz) stability of $C_6D_6^+$ during flights <0.3% and therefore had practically no effect on signal from ambient $NH_3$. Occasional slower drifts did occur, typically changes in primary ion signal of up to 13% over 1-3 min. We hypothesize that those drifts were related to temperature changes of the benzene reservoir, as

they sometimes appeared to coincide with ~2-K drifts in cabin air temperature. There was, unfortunately, no temperature measurement at the benzene reservoir. We would generally recommend controlling the benzene reservoir's temperature, although we did not do so. An additional observation was that the primary ion signal often started off about 15% to 18% low upon switching from negative to positive polarity. It would take ~10 min to reach a stable value of 2-3 $\times$ $10^6$ cps. We speculate that behavior was due to the re-stabilization of ion guidance elements in the atmospheric-pressure interface of the

mass spectrometer.

Normalized count rates of $NH_3.C_6D_6^+$ were then divided by calculated sensitivity values. We estimated a maximum (high-RH) sensitivity of 4.4 ncps/pptv and multiplied it by the relative sensitivity as a function of IMR water pressures (orange in Fig. 2) that were calculated from temperature and humidity measurements throughout each flight. Instrument background signals were determined by linearly interpolating between the short and frequent zero measurements and subtracted from the

signals observed during ambient sampling, thus obtaining the ambient $NH_3$ mixing ratios. We could not rigorously establish the uncertainty in used sensitivities (see also Sect. 3.8), but a conservative estimate of ±1 ncps/pptv would result in a systematic error for the reported mixing ratios of typically 20% to 30%.

Figure 4 illustrates the process of obtaining $NH_3$ mixing ratios for a segment of RF6 that featured variations of mixing ratios as they were typically encountered. For most of that segment, the aircraft was flying within the boundary layer and we

observed $NH_3$ levels between 3 and 5 ppbv. As also presented in Fig. 3, dry in-flight zeros dropped count rates by about two orders of magnitude, though about one order was attributed to momentarily decreased sensitivities. Between 17:00 and 17:15 (UTC), the aircraft made an excursion into the free troposphere, and $NH_3$ mixing ratios dropped down to ~1 ppbv. After 17:30, the aircraft twice crossed a plume, during which we observed $NH_3$ mixing ratios up to >30 ppbv. The background

counts also responded to such transitions in ambient NH₃ levels. But even in the sharp and drastic plumes, that response was
subdued, as expected (cf. Fig. 3), and the immediate instrument response times thus hardly affected.

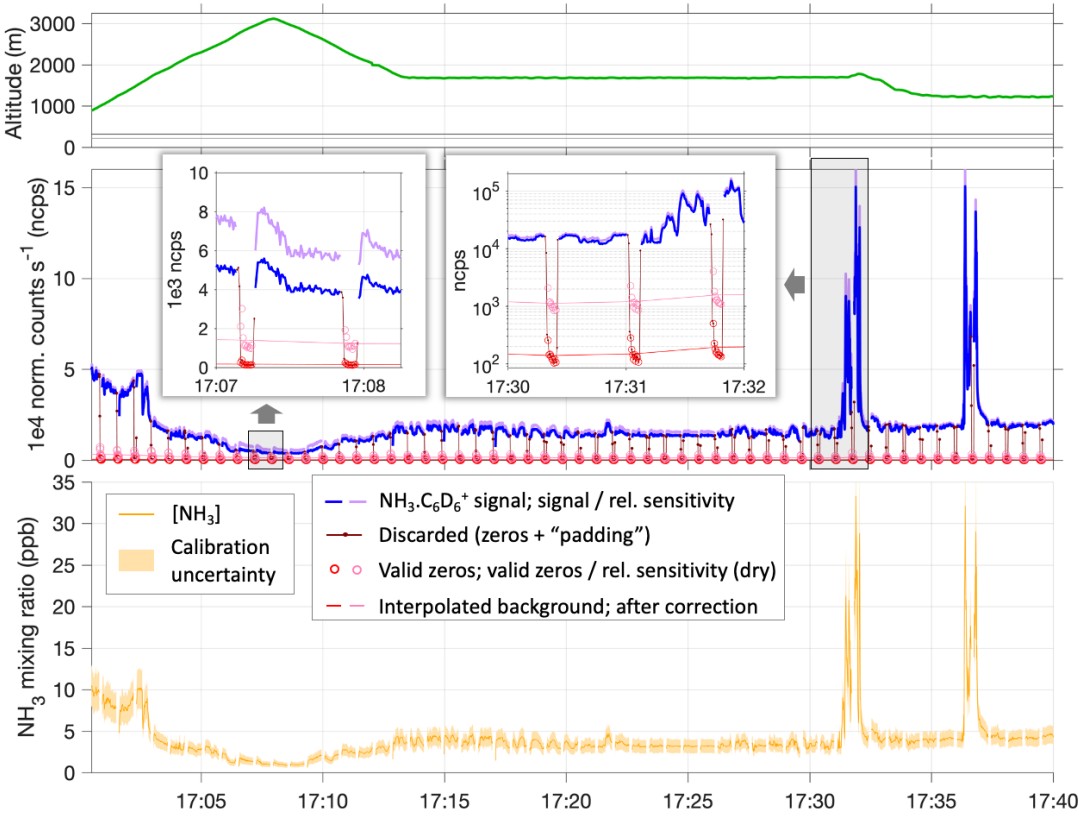

**Figure 4: Example time series showing instrument time response to NH₃ in plumes and instrument zeros, based on a segment of RF6 on May 3. The top panel shows the flown altitude profile in meters above mean sea level; horizontal lines marking the elevations of the SGP ground site (dark gray) and KBVO airport (light gray). The central panel shows normalized count rates of NH₃·C₆D₆⁺ (blue, in-flight zeros in red). Sensitivities as per calibration results were applied (cf. Fig. 2), including consideration of their humidity dependence (pink and purple). Insets highlight the response to plumes of high NH₃ levels, to measurements in the much cleaner and drier free troposphere, and the respective in-flight zero measurements. Background signals were interpolated between zero measurements (red and pink lines) and subtracted from the ambient signals to obtain ambient NH₃ mixing ratios (bottom panel). The sensitivity values used here range from 2.8 to 4.2 ncps/pptv, with a general uncertainty of ±1 ncps/pptv estimated based on the calibration tests (indicated by shades in bottom panel).**

Overall, we could conclude that the initially fast instrument response, on the order of a second, makes the instrument very well-suited for airborne in-situ measurements of NH₃ mixing ratios. The remaining instrument background responded more slowly. Throughout all flights, the corrected backgrounds remained low enough so that their subtraction did not incur significant uncertainties, but they might become an issue during sufficiently drastic transitions to very high or in particular to very low NH₃ mixing ratios. Correspondingly, mixing ratios in plumes may be overestimating, as we did not correct for slow background response (see next session for an estimate). Conversely, some of the mixing ratios we report here for the lower free troposphere may be underestimating when measured during a climb out of the boundary layer. However, [NH₃] only

dropped by a factor of more than 5 on one such occasion, specifically during RF8 (May 7), with a drop of about an order of magnitude (Fig. S3). If we pessimistically assumed the (humidity-corrected) background signal was one order of magnitude too high for the free tropospheric measurements in this case, the reported mixing ratios of ~200 pptv would be about 25% too low in this worst-case scenario.

The flight chosen for Fig. 4, RF6, featured both the highest $NH_3$ mixing ratios observed during HI-SCALE's IOP1 and the highest $NH_3$ levels on average. An overview of all the mixing ratios obtained during 11 research flights in May 2016 is given in the supplement, with results presented in Fig. S3. In short, the observed mixing ratios spanned more than 2 orders of magnitude, from 100 pptv to tens of ppbv. Mixing ratios < 1 ppbv were measured either during cloudy days and above cloud base, or clearly in the free troposphere, whereas $NH_3$ mixing ratios > 1 ppbv were observed on overall sunny days, clear of cloud and with good confidence also within the boundary layer, within which $NH_3$ appeared vertically well mixed. Climbing into the (lower) free troposphere on these flights, mixing ratios dropped by factors of 3 to 30.

### 3.4 Fertilizer plant plume transects

As mentioned above, and seen in Fig. 4, RF6 repeatedly crossed a plume of substantially enhanced $NH_3$ levels. In fact, the same plume was fully crossed five times during that flight, at altitudes between 500 and 1700 m AMSL (Fig. 5a). Given the prevailing southwesterly winds at the time and the locations of the plume crossings, a large fertilizer plant (Koch Fertilizer LLC) was identified as the $NH_3$ source, located 35 km SW from the SGP site, near the town of Enid, OK (Fig. 5b). The US Geological Survey's 2016 Minerals Yearbook ranks this plant as the 4th-largest domestic producer of anhydrous $NH_3$, with a production capacity of 930 000 tons per year.

To estimate the observed source rate, we crudely estimated the total amount of horizontally advected $NH_3$ (*M*) for each plume transected, based on the assumption that the plume filled out the full boundary layer in the vertical. For previous instances of that simple mass balance approach, see, e.g., Turnbull et al. (2011) and references therein. The altitude of the boundary layer top was estimated based on the characteristic drops in humidity (1700 m around 17:30, 2000 m around 19:00, with a conservative uncertainty of ±200 m due to spatio-temporal variabilities). The CIMS measurements provided estimates for the experienced widths of the plumes (*w*), and plume and background concentrations of $NH_3$ ($c_p$ and $c_{bg}$). The cross-wind component was then used to calculate the horizontal $NH_3$ mass flux (*M*) perpendicular to the plume cross-section:

$$M = \left( c_p - c_{bg} \right) \cdot h \cdot w \cdot V \cdot |\sin \theta| \tag{Eq. 1}$$

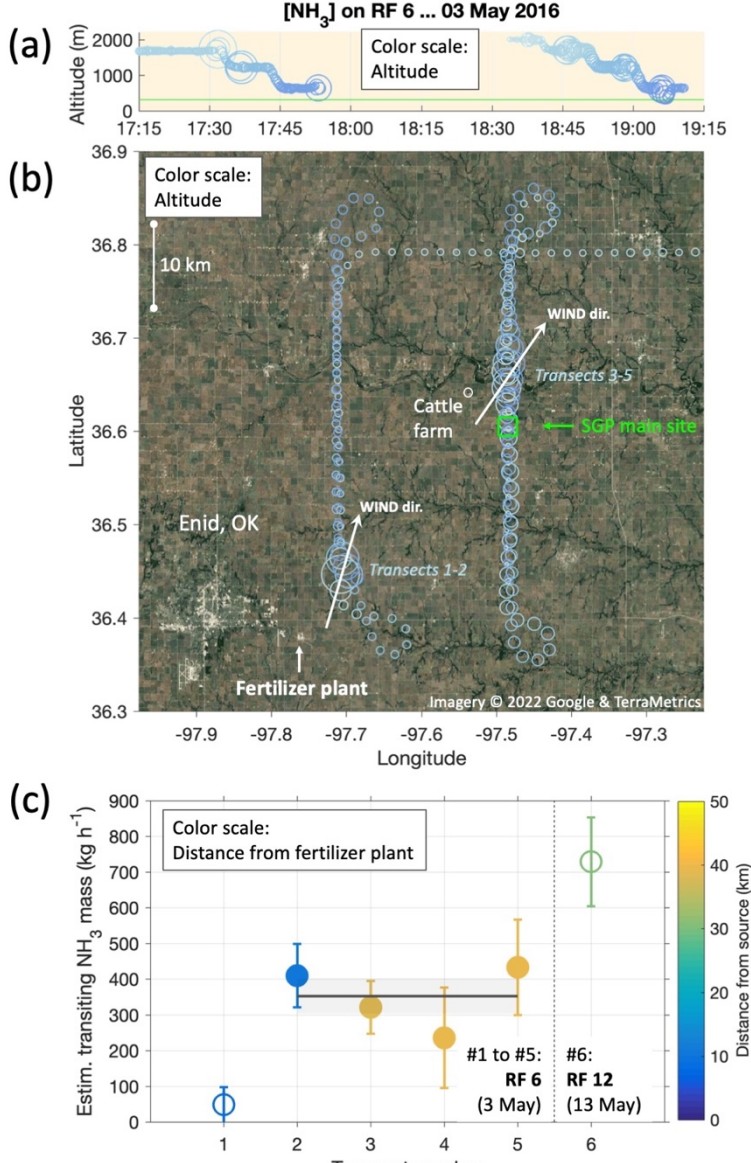

**Figure 5: Source rate estimation for a large fertilizer plant (Koch Nitrogen LLC) in the town of Enid, OK, based on horizontal fluxes calculated from plume transects during RF6 on 3 May. Panel (a) shows the altitude profiles flown from 17:15 to 19:15; panel (b) the corresponding ground tracks in relation to Enid, the fertilizer plant and the SGP ground site (blue square). Circles sizes correspond to measured NH₃ mixing ratios; colors correspond to flown altitudes. Arrows in panel (b) indicate the average wind directions for respectively co-located plume transects 1-2 and 3-5. Panel (c) presents the mass of NH₃ transiting in the total plume, estimated as per on Eq. 1. The weighted mean using the four most reliable transects amounts to 350 ± 50 kg h⁻¹ (gray line and shades). Also shown is a plume transect (#6) during RF12 on 13 May, for which the same calculation yielded 730 ± 125 kg h⁻¹.**

In Eq. 1, $h$ is the boundary layer depth, and $\theta$ the angle between the wind direction and the aircraft ground track while transecting the plume. Figure 5c shows the resulting amounts of transiting $NH_3$, expressed in kg h⁻¹, for the five plume transects in RF6, and in addition for an additional transect of a plume from the same plant during RF12 ten days later. We

discounted the first transect, which gave a much lower advection rate than the other transects, probably due to the proximity to both the plant and the boundary layer top, suggesting insufficient mixing. It also yielded a high relative uncertainty due to a very small $\theta$. On average, transects 2-5 yielded a source rate of $M = 350 \pm 50$ kg h$^{-1}$. We had neglected here the response time of the instrument backgrounds, which our experiments and observations suggested would occur on timescales longer than the plume transects (Figs. 3-4). The ensuing error would be an overestimation, though even at worst by less than 10%.

In any case, our result for $M$ is of the order broadly expected, as the US Environmental Protection Agency's (EPA) 2017 National Emissions Inventory (NEI) lists the Koch site at Enid as emitting a total of 1 905 tons of NH$_3$ per year, which corresponds to an average of 218 kg h$^{-1}$.

### 3.5 Case flight for eddy covariance analysis: RF13 (14 May 2016)

The HI-SCALE airborne campaign focused on aerosol-cloud interactions, and its flight profiles were not designed for
quantifying emission fluxes. Furthermore, the CIMS mostly did not operate in the benzene-CI mode that allowed for detection of NH$_3$. Therefore, we focus here on one research flight, RF13 (14 May), which provided the most suitable dataset for analyzing eddy covariance (EC) fluxes of NH$_3$, with three straight-and-level legs within the turbulently mixed boundary layer, for > 10 min each. Our analysis primarily showcases the capability of the setup to derive airborne EC fluxes, and we explore the suitability of our datasets for ensemble average (EA) and continuous wavelet transform (CWT) flux calculation
methods.

The three example legs of RF13 occurred in the early afternoon of 14 May 2016. Surface temperatures were between 12 and 16 °C with RH between 36% and 39%. Conditions were generally sunny, with few, occasionally scattered clouds with a base above 3000 m AMSL. There was a marked drop in both humidity and NH$_3$ (Fig. 6b), along with a temperature inversion, at 1450 m AMSL (19:27) and later at 1650 m AMSL (20:08), likely marking the top of the turbulently mixed boundary layer.
All three example legs overflew the SGP site around their mid-points. The first leg was flown between 18:10 to 18:25, at 320 m above ground, from NE to SW; the second between 19:25 and 19:40, at 580 m above ground, from NW to SE; the third between 19:40 and 19:55, at 310 m above ground, from SE to NW. Boundary-layer mixing ratios of NH$_3$ were mostly between 1.5 and 2 ppbv (Fig. 6b), or ~1.2 µg m$^{-3}$. The on-board aerosol mass spectrometer observed sub-micron aerosol loadings between 0.9 and 1.8 µg m$^{-3}$, thereof 0.16 to 0.26 µg m$^{-3}$ of particulate ammonium ($p$NH$_4$), yielding a gas-to-particle
partitioning ratio of ~6:1. In the free troposphere, that ratio decreased, as [NH$_3$] dropped to 400 pptv (0.24 µg m$^{-3}$) but $p$NH$_4$ only to 0.1 µg m$^{-3}$ (Figs. 6b and 6c). This observation may imply that the sub-micron aerosol was more acidic in the free troposphere than in the boundary layer (Pye et al., 2020). Figure 6 also indicates that particulate NO$_3$ and $p$NH$_4$ concentrations within the boundary layer tended to increase with altitude, despite broadly constant availability of NH$_3$, which is consistent with lower temperatures favoring NH$_4$NO$_3$ formation.
We also found that our measured NH$_3$ levels agreed well with Weather Research and Forecasting model (WRF-Chem) predictions. The model was configured to cover a domain extending > 1000 km from the SGP site in every direction at a grid spacing of 12 km. For the flight track and times of RF13, WRF-Chem predicted between 1.2 and 1.4 ppbv for the boundary

layer, falling within the range of 1.1 to 2.4 ppbv we measured (Fig. 6b). Further details on the WRF-Chem model configuration, including NH₃ emissions, are given in the supplement.

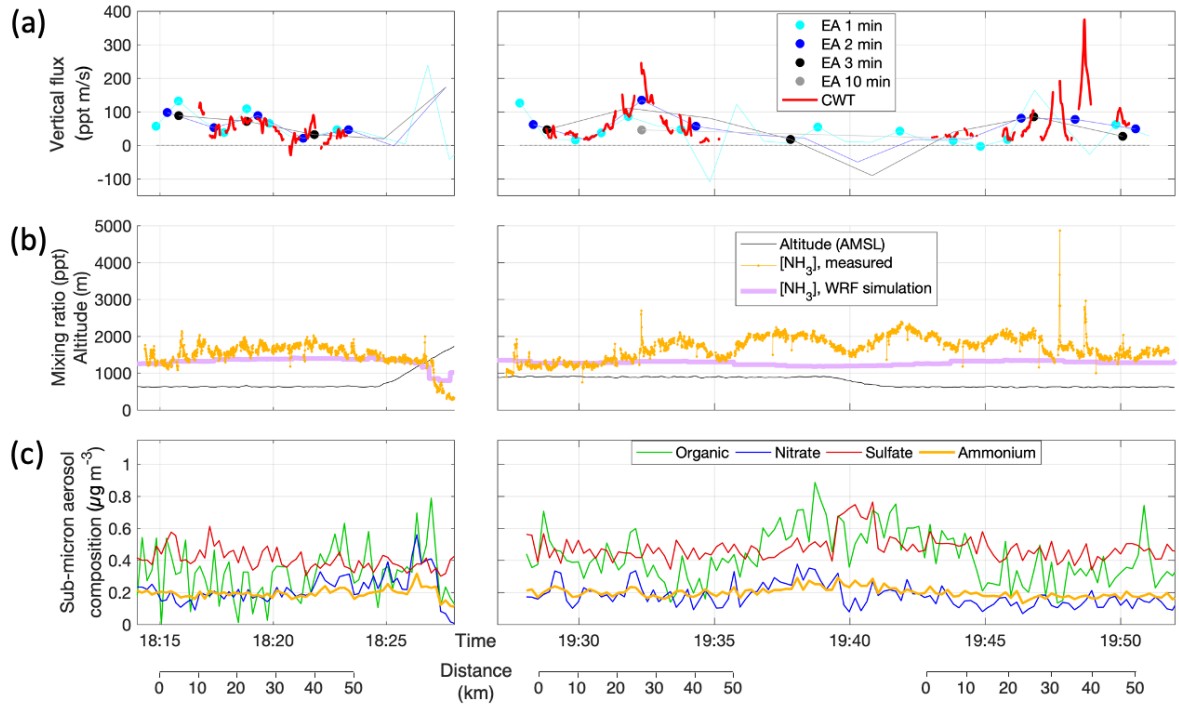

**Figure 6: Top (a): Eddy-covariance NH₃ flux calculation results for three legs within the mixed boundary layer during RF13 on 14 May. The first and third leg were flown at 310-320 m above ground; the second leg at 580 m and directly above the third one but in opposite direction. Thin lines without markers connect all results using the ensemble-average method (EA); circles mark those results that fulfilled the quality criteria (stationarity test, lag correlation, altitude stability). EA fluxes were determined for time windows of lengths 1 (cyan), 2 (blue), 3 (black) or 10 min (gray), ±50% each. Note that straight-and-level legs were required for obtaining sensible values for EA fluxes, leading to erratic results during climbs (after 18:25) and descents (~19:40) that did not pass quality checks. Center (b): NH₃ mixing ratios as measured (orange) and as in WRF model simulation (purple) along the flight track, together with flight altitude (black), using the same ordinate in units of pptv and m AMSL, respectively. Bottom (c): Contributions of nitrate (NO₃), sulfate (SO₄), ammonium (NH₄) and non-refractory organic material to the mass concentration of sub-micron aerosol particles, as measured by the on-board aerosol mass spectrometer (AMS).**

### 3.6 Eddy covariance analysis

For the EC analysis, we used a custom-made MATLAB toolbox (Wolfe, 2022) that was also used for airborne EC in Wolfe et al. (2018; hereafter referred to as W2018). Only minor modifications were necessary to adapt the scripts to our dataset. We refer to W2018 also for a more comprehensive discussion of the involved flux methodology, which we largely followed here.

In the traditional EA method, individual flux values are calculated for a pre-defined time interval each. That flux, $F_{EA}$, is simply the covariance of the timeseries for a scalar $s$ (here, NH₃ mixing ratio) and the vertical wind speed $w$ during that time interval. We calculated $F_{EA}$ for a variety of interval lengths (from 1 to 10 min), for assessing the general feasibility of the EA method for obtaining EC fluxes from our dataset. The results are presented in Fig. 6a (blues and grays); lines connecting all

results, markers being used only if quality checks were passed, in particular the stationarity test (Foken and Wichura, 1996; using three sub-intervals and requiring <35% deviation). The results are internally consistent, as $F_{EA}$ using shorter time intervals (down to 1 min, corresponding to ~6 km, cyan) broadly average to the results using longer intervals. This suggests that little flux was 'lost' even when limiting the covariance calculation to successions of as small as 1-min intervals. Smaller intervals would also generally increase the chances of passing the stationarity test. They here corresponded to a spatial

resolution of 6 km along the flight track. More details on this analysis are provided in the supplement, with Fig. S4 showing lag correlations, co-spectra, and power spectra for $NH_3$ and temperature data. The co-spectra confirm that most flux was indeed carried by eddies observed at periods shorter than 1 min (i.e., frequencies > 0.017 Hz). The power spectra exhibit the $f^{-5/3}$ power dependence towards high frequencies, as theoretically predicted for the inertial subrange (Kaimal and Finnigan, 1994), and suggesting that high-frequency attenuation was negligible up to the Nyquist frequencies (1 Hz for the $NH_3$ data).

In the CWT method, covariance is analyzed via the continuous wavelet transforms of $w$ and $s$, $W_w$ and $W_s$, which are the convolutions of their time series with scaled and translated versions of a time-dependent 'mother wavelet' function (Torrence and Compo, 1998; Mauder et al., 2007; W2018). The wavelet 'cross-scalogram', $W_w W_s^*$ (* denoting the complex conjugate), is a function of scale (frequency) and translation (time) and its real part (times a conversion factor) corresponds to local co-spectra for each point in time (e.g., Fig. 7b). Their scale-weighted sum over all scales yields a time series for the

covariance between $w$ and $s$, i.e., flux ($F_{CWT}$). The CWT method has important advantages over the traditional EA method, especially when it comes to flux calculations using aircraft data. A major advantage is that stationarity is not required. There is hence also generally no need for detrending the input time series or dividing them into intervals. Instead, one obtains a continuous time series of fluxes, along with time-resolved contributions of scales. The consequent (at least theoretically) high time resolution does not come at the expense of neglecting lower-frequency contributions, which is an inherent tradeoff

when going for higher time resolutions using the EA method. For airborne measurements in particular, the higher time resolution corresponds to a finer spatial resolution, and by not relying on stationarity, heterogeneous conditions (e.g., due to heterogeneous surface emissions) can be investigated more readily. Due to these advantages, CWT has been applied for calculating EC fluxes from airborne measurements for decades (e.g., Attié and Durand, 2003; Mauder et al., 2007; Karl et al., 2009).

In our CWT flux analysis, we used the Morlet wavelet with a wavenumber of 6, the standard choice for eddy covariance applications (Schaller et al., 2017). We applied lag time as obtained by the EA flux calculations (typically < 1 s; zero lag was used for RF13; see Fig. S4). Figure 7 presents the fluctuations of the $w$ and $s$ = [$NH_3$] time series, local co-spectra and resulting $F_{CWT}$ time series for the three selected legs. The regular gaps in the data (e.g., white stripes in Fig. 7b) are due to the frequent background determinations in the CIMS measurement routine. We dealt with these data gaps by filling them with

covariance-based projected values as suggested in W2018 ('covariance filling'). In agreement with their work, this method led to apparently smaller artifacts in the vicinity of the gaps than other gap-filling methods. To be conservative, we anyway discarded results for within the gaps as well as half a gap width on either side, which appeared sufficient even if we instead used the more artifact-prone 'stitching' method (which simply removes the times of the gaps, 'stitching' the time series

together). The resulting gaps for the flux time series were 15 s wide and occurred every 42 s. Shades in Fig. 7b also illustrate the 'cone of influence' (COI) for each leg, which refers to the scales and locations of the wavelet coefficients, and hence co-spectra, that could be influenced by data that remained unmeasured before and after the leg. Co-spectral power within the COI may thus be subject to edge artifacts. When we calculated $F_{CWT}$, we included the COI (orange line in Fig. 7c). But to be conservative again, we discarded any fluxes for which the COI extended to periods < 60 s, or for which the flux excluding the COI (thin black line in Fig. 7c) differed by > 50%. Altogether, almost two thirds (65%) of the full $F_{CWT}$ time series were thus discarded: ~35% due to the 15-s gaps, and another ~30% due to the COI filtering.

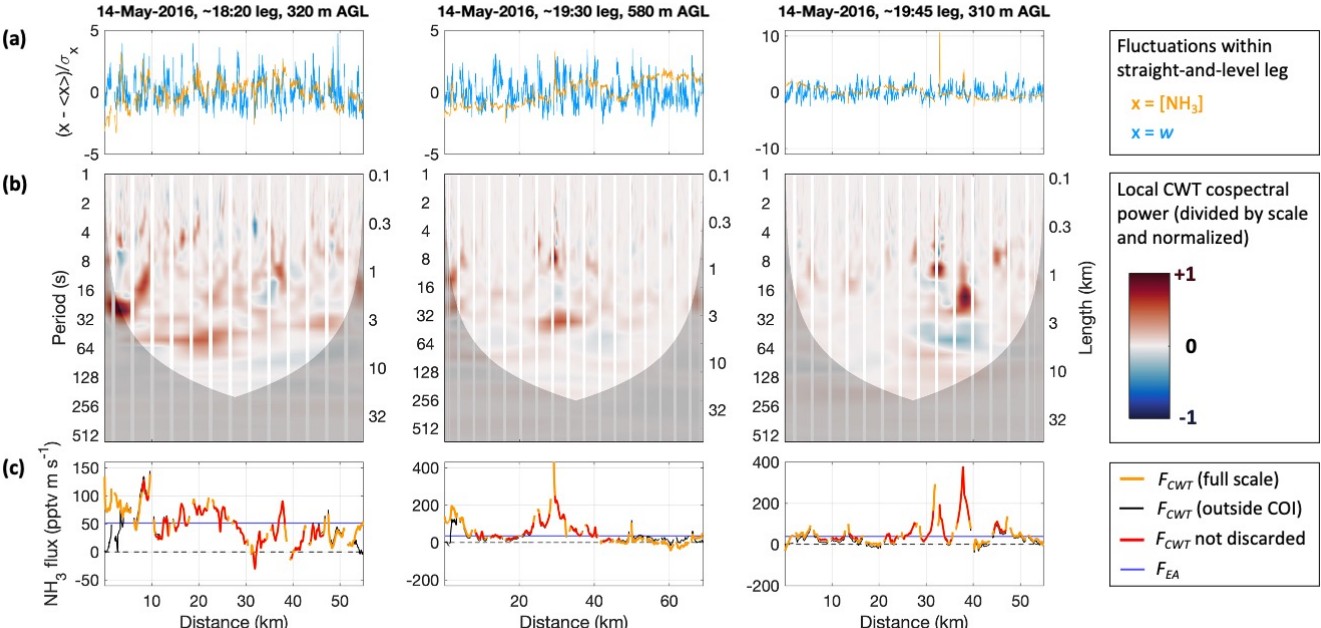

**Figure 7: Continuous wavelet transform (CWT) analysis of the covariance between vertical wind ($w$) and NH$_3$ mixing ratio ([NH$_3$]) for the three selected straight-and-level legs in three columns. Panels in each column share the same abscissa, representing distance along the respective leg. The regular gaps in the data are the zero measurements by the CIMS. Top panels (a) show the normalized time series, i.e., fluctuations around the leg's mean. Panels (b) present local co-spectral powers as surface plots, following scale bias correction and normalization; reds for positive (upward) power, blues for negative (downward) power. The darkened lower parts are the 'cones of influence' (COI) that mark the locations and scales where co-spectral power may be subject to edge effects. Scales are expressed as periods (left ordinate) and lengths (right ordinate); periods correspond to lengths as per the aircraft's ground speed (90-98 m s$^{-1}$). Bottom panels (c) show the resulting flux values, $F_{CWT}$, orange for the full-scale averages, thin black for averages outside the COI only, and red for the data passing the conservatively chosen quality criteria. For comparison the ensemble-average covariance, $F_{EA}$, is shown in blue.**

The obtained $F_{CWT}$ values compared well overall to the fluxes obtained using the EA method (Fig. 6a), in particular for $F_{EA}$ in 1-min intervals. Using such small intervals, the EA method allowed for retrieving fluxes closer to the edges of each leg. However, that was achieved by excluding any larger-scale fluctuations and covariance *a priori*, which the CWT method did not. In addition, the CWT procedure acknowledged the possible but unknown influence of larger scales towards the leg perimeters, via the COI considerations above. Further away from the leg perimeters, fewer $F_{CWT}$ were flagged due to COI. And despite the zeroing gaps, the CWT method clearly achieved a denser coverage here than the EA method, mainly because

it did not rely on stationarity, while NH$_3$ mixing ratios would experience both gradual changes as well as several sharp plumes of varying intensity. In particular in the presence of strong plumes, the $F_{EA}$ failed their quality checks, whereas the $F_{CWT}$ time series responded with peaks on their own, which we will further explore below. Nominally, the CWT method yielded fluxes at the frequency of the scalar measurements (2 and 20 Hz), but the "true" time resolution of the fluxes is of course much lower, which was reflected by the wider peaks in $F_{CWT}$ as compared to the corresponding peaks in the [NH$_3$] time series.

The co-spectral powers shown in Fig. 7b include a so-called bias correction for wavelet scale (Liu et al., 2007), as performed also for calculating the EC fluxes. With that, they illustrate that most of the power contributing to $F_{CWT}$ was at scales smaller than 1 min. This is apparent also from the leg-wide averages, or analogously from the leg-wide frequency-weighted Fourier transform co-spectra (Fig. S6; quality-controlled locations only), and in agreement with the co-spectra obtained following the EA method (cf. Fig. S4, center row). Likewise, the wavelet power spectra of the measured time series (Fig. S7) were similar to their Fourier transform counterparts (cf. Fig. S4, bottom row). Figure S7 shows wavelet power spectra also for $w$ and ambient temperature ($T$) measurements. Turbulence was captured well by the $w$ measurements up to highest frequencies, whereas the Rosemount's 1-Hz $T$ measurements were somewhat attenuated at frequencies > 0.3 Hz. For further discussion of the $T$ power spectra, see the supplement.

**3.7 Vertical flux divergence**

The motivation of EC flux measurements is often the investigation of atmosphere-surface interactions. If certain conditions are fulfilled, the measured EC fluxes correspond to net emissions from or net deposition to the surface, or they can be used to calculate that net air-surface exchange. With rare exceptions (Crawford et al., 1996; Sayres et al., 2017), it is not feasible to perform research flights within the shallow near-surface 'constant flux layer', which extents roughly up to 10% of the mixed layer depth. Above that layer, the flux typically decreases with altitude ($z$), and this vertical flux divergence needs to be considered for typical airborne EC measurements. In general, flux divergence $\partial F/\partial z$ for a scalar $s$ can result from several processes (W2018):

$$\frac{\partial F}{\partial z} = -\frac{\partial s}{\partial t} - \bar{U}\frac{\partial s}{\partial x} + Q \qquad \text{(Eq. 2)}$$

where the first term on the right side is storage, and the second horizontal advection, with $\bar{U}$ the horizontal wind speed in direction $x$, and the last term the local net source or sink. Subsidence and horizontal turbulent terms have been neglected in Eq. 2, as they are typically at least an order of magnitude smaller (Karl et al., 2013). Our flight profiles did not allow us to assess the terms on the right side of Eq. 2. However, for scalars with no or only slow atmospheric sources or sinks (e.g., non-reactive species), the flux divergence is expected to be linear throughout most of the boundary layer (Vinuesa and Arellano, 2011). Once could expect such linearity also for NH$_3$, as its boundary-layer lifetime against oxidation is weeks to months (Diau et al., 1990), while its gas-particle partitioning can be assumed to be in equilibrium, at least outside plumes. Flux divergence may then be obtained more directly by measuring fluxes at multiple altitudes, as suggested, e.g., in W2018. For

species with vertically inhomogeneous source or sink rates, however, flux divergence may be non-linear (Wolfe et al., 2015). Indeed, the partitioning of $NH_3$ into the particle phase is expected to be enhanced at lower temperature, which generally decreases with height within the boundary layer (Sect. 3.5, Fig. 6). In any case, we were unfortunately unable to consistently investigate flux divergence for this study, as the only pair of suitable legs (vertically stacked, horizontally co-located and close in time) were the 2nd and 3rd legs of RF13, as shown in Fig. 6, and much of the respective flux time series was riddled with gaps and subject to near-by point sources. For the remaining pairs of flux values, we obtained a median $NH_3$ flux divergence of –0.02 pptv $s^{-1}$, although with a standard deviation of 0.15 pptv $s^{-1}$, i.e., at insufficient accuracy and precision. That divergence rate is, however, of the expected order, as it corresponds to ~1 mol $km^{-2}$ $h^{-1}$ per 315 m of height or to a loss of ~15% compared to typically measured fluxes at that height. To simplify the remainder of our footprint analysis, we continued with using data obtained from 315 m AGL and neglected any, presumably slightly low, bias due to flux divergence.

### 3.8 Flux uncertainties

We investigated the uncertainties in our flux calculations following primarily procedures as outlined in detail in W2018, extended by some considerations specific to our case. We start with a compilation of various sources of systematic errors for mixing ratios and fluxes, followed by a discussion of random flux errors. Table 1 summarizes the results for each of the three legs in RF13. In the following, "typical" ranges refer to interquartile ranges.

The flux errors directly propagate into surface exchange rates, and the scaling to account for flux divergence can introduce additional uncertainty (W2018). However, our limited datasets did not allow us to assess these quantities (Sect. 3.7).

### 3.8.1 Systematic errors

The largest systematic error was likely due to the limited accuracy of the $NH_3$ mixing ratio measurements ($SE_{acc,MR}$), caused by the uncertainty of the sensitivity values used to convert count rates to mixing ratios (Sect. 3.1 and 3.3). We could only crudely estimate that uncertainty. For the maximum (dry) sensitivity of 4.4 ncps/pptv, an uncertainty of ± 0.2 ncps/pptv would correspond to a relative systematic uncertainty of 5% – likely an optimistic estimate for overall accuracy, given its humidity dependence (Fig. 2). If we changed to the steeper humidity dependence found later (blue in Fig. 2; instead of orange), $NH_3$ mixing ratios would typically increase by 23% to 26% – likely a pessimistic estimate for the instrument's accuracy. $SE_{acc,MR}$ is of unknown sign and would usually propagate, in relative terms, directly to the derived fluxes. $NH_3$ flux, however, could also be affected by small fluctuations in ambient water vapor, which itself had a consistent upward flux (as typical for the turbulent boundary layer due to evaporation from the surface). These fluctuations were in principle accounted for, as 1-Hz humidity data were used for calculating $NH_3$ mixing ratios, but uncertainty in the humidity dependence could potentially lead to larger errors for $NH_3$ fluxes, $SE_{acc,F}$ (e.g., Fig. S8). Using the same procedure as above, we obtained a high estimate for $SE_{acc,F}$ of 10% to 28%, again of unknown sign. The variability of these upper estimates for $SE_{acc,F}$ is illustrated in Fig. S9.

Other systematic error sources for fluxes are under-sampling of turbulent fluctuations at low as well as high frequencies. These errors correspond to low biases in the absolute values of the measured fluxes. For airborne CWT fluxes, sampling of low frequencies is primarily limited by the finite length of the flight leg. An upper limit for the resulting fractional systematic error, $SE_{turb}$, can be estimated as a function of leg length, flight altitude and boundary layer depth (Lenschow et al., 1980; W2018), yielding between 2% and 3%. High-frequency sampling is mainly limited by the response time of the instrument. Using the worst-case response time of 0.4 s (Sect. 3.2), integration over transfer function-weighted leg-wide co-spectra (Horst, 1997; W2018) yielded systematic error fractions, $SE_{RT}$, of 4% to 12%. Additional under-sampling of high frequencies may have occurred as the CIMS data were acquired at only 2 Hz, whereas the standard for EC is 10 Hz. However, the co-spectra (Figs. S4, S6) show quickly diminishing flux contributions as frequency increases towards 1 Hz. The corresponding ogives (not shown) indicate that 92% to 96% of co-spectral power occurred at frequencies < 0.2 Hz, which is a fifth of our Nyquist frequency of 1 Hz. With that, we crudely and conservatively estimate that quintupling our sampling rate to 10 Hz would add at most 4% to 8% of flux ($SE_{SR}$).

Table 1: Estimates for systematic errors (SE) and random errors (RE) for the case study of flight RF13. Descriptions for each error type are given in the text (Sect. 3.8). For SE, signs indicate error direction relative to the absolute values of measured fluxes; '±' means the direction could be either side. Ranges of errors, indicated by '…', correspond to leg-wide interquartile ranges; errors without ranges apply to each flux in the leg.

| Error type | Applying to | Leg 1 (320 m AGL, 55 km) | Leg 2 (580 m AGL, 69 km) | Leg 3 (310 m AGL, 55 km) |
|---|---|---|---|---|
| $SE_{acc,MR}$ | Mixing ratios | ± 5% to ± 22…25% | ± 5% to ± 22…26% | ± 5% to ± 24…27% |
| $SE_{acc,F}$ | Fluxes, emissions | ± 5% to ± 6…20% | ± 5% to ± 19…31% | ± 5% to ± 11…30% |
| $SE_{turb}$ | " | + ≤ 2% | + ≤ 3% | + ≤ 3% |
| $SE_{RT}$ | " | + 8% | + 12% | + 4% |
| $SE_{SR}$ | " | + < 8% | + < 6% | + < 4% |
| $RE_{wave}$ (0.1 Hz) | " | 70…161% | 86…140% | 74…147% |
| $RE_{turb}$ (leg) | " | ≤ 18% | ≤ 20% | ≤ 19% |
| $RE_{noise}$ (leg) | " | ~18% | ~17% | ~16% |
| $(RE_{turb}^2 + RE_{noise}^2)^{1/2}$ | " | ≤ 26% | ≤ 26% | ≤ 25% |
| $RE_{wave}$ (leg) | " | 32% | 26% | 12% |
| Flux divergence | Emissions | not determined | not determined | not determined |

### 3.8.2 Random flux error

For assessing total random flux error, we followed the method proposed in W2018. It uses the wavelet coefficients of the measured scalar (i.e., [$NH_3$]) and vertical wind to calculate cross- and auto-covariances across a range of lag times. Total random error ($RE_{wave}$) is then obtained empirically via estimating the variance of the covariance over of a certain range of

lags (Finkelstein and Sims, 2001). We used lags ± 10 s, so the range would capture the integral timescale (~3-7 s; Fig. S5) while keeping the contribution of trends in the time series low. Further following previous works (Mauder et al., 2013; W2018), we did not consider frequencies lower than $f_{min}$ = 0.02 Hz (spatial scales > ~5 km), again to limit the potential influence of trends (cf. Figs. 7, S6). For 1-Hz fluxes, the resulting typical $RE_{wave}$ were 250% to 490%; averaging to 0.1 Hz fluxes (spatial scales of ~1 km) yielded 'more useful' $RE_{wave}$ of typically 79% to 145% (Fig. 8; Table 1). Note that the choice of the low-frequency cut-off ($f_{min}$) had a marked influence on that result: the median $RE_{wave}$ of 116% for $f_{min}$ = 0.02 Hz would increase to 156% for $f_{min}$ = 0.01 Hz, or decrease to 73% for $f_{min}$ = 0.04 Hz, or 34% for $f_{min}$ = 0.1 Hz. Total random flux error is due to inherent randomness of boundary-layer turbulence ($RE_{turb}$) and uncorrelated instrumental noise ($RE_{noise}$). Theoretical upper bounds to $RE_{turb}$ were estimated leg-wise (Lenschow et al., 1994), similar to $SE_{turb}$, yielding ~19%. Noise in the [$NH_3$] time series was estimated from lagged auto-covariances (Langford et al., 2015), yielding $RE_{noise}$ of ~17%. The estimated total random errors were thus ~26% (or less, as $RE_{turb}$ are estimated upper bounds), which broadly agreed with the leg-averaged $RE_{wave}$ of 12% to 32% (Table 1). Analogously to W2018, we thus concluded that the calculated $RE_{wave}$ were of the correct order.

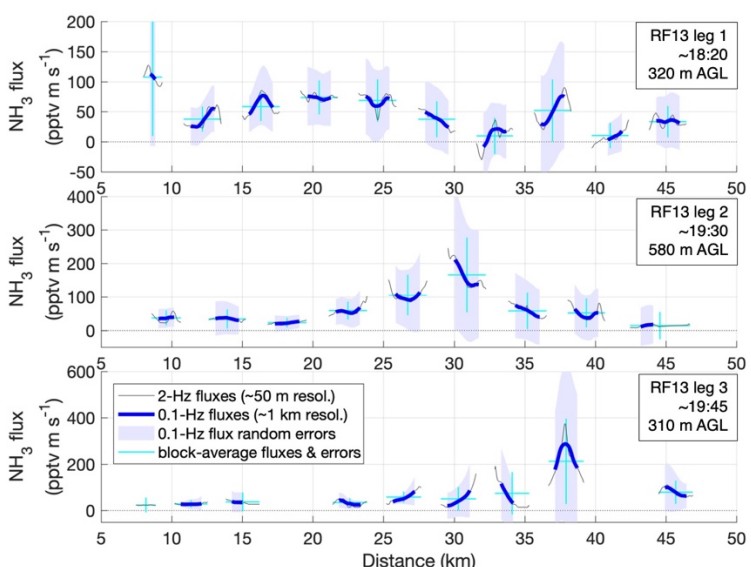

**Figure 8: Total random flux errors along the $F_{CWT}$ time series for the three case-study legs (Figs. 6, 7), estimated empirically using wavelet coefficients ($RE_{wave}$). Quality-controlled $F_{CWT}$ (red in Figs. 6, 7) are shown (thin gray lines), along with moving averages to 0.1 Hz, corresponding to spatial scales of ~1 km (thick blue lines). Shades are $RE_{wave}$ (± 1 σ) for those averages. Horizontal cyan bars mark the width of blocks of continuous $F_{CWT}$ (~2.5 km) and their block-wide averages; vertical cyan bars denote the $RE_{wave}$ for these averages.**

For spatially resolved $F_{CWT}$ values, this total random error was clearly the dominant uncertainty. It is illustrated in detail in Fig. 8 for the three legs in RF13, and it will be used as the uncertainty estimate for fluxes and net surface exchange in the remainder of this chapter. For simplicity, none of the systematic flux errors are henceforth considered, including those of known sign, which would amount to slight increases in absolute flux values.

## 3.9 Flux footprint analysis

The flux footprint is the area of the surface that contributes to the net flux observed at a certain location and height. The footprint is a generally 2-dimensional function of location and describes how strongly sources and sinks in the area contribute to the flux. For our study, we calculated flux footprints based on relatively simple parametrizations. Weil and Horst (1992) proposed as a metric the half-width of the horizontal (1-dimensional) footprint, $dx_{0.5}$:

$$dx_{0.5} = 0.9 \frac{\bar{U} z_m^{2/3} h^{1/3}}{w^*} \tag{Eq. 3}$$

Like above, $h$ is the mixed layer depth and $\bar{U}$ the horizontal wind speed; $z_m$ is the measurement height, and $w^*$ the Deardorff convective velocity scale, typically in the range of 1-2 m s$^{-1}$ (Stull, 1992; Karl et al., 2013). We estimated $h$ like above (1100-1300 m AGL), and calculated $w^*$ based on our best estimate for the sensible temperature flux during RF13 (0.16 K m s$^{-1}$) yielding ~1.9 m s$^{-1}$. We obtained the *shape* of the 1-D footprint functions from the crosswind-integrated footprint predictions that resulted from the parametrization presented in Kljun et al. (2015), which itself is based on Lagrangian stochastic particle dispersion simulations. The predicted shapes are identical when referenced to the horizontal distance of the footprint distribution's median from the measurement location, at least for the range of conditions we encountered. The footprint shape is presented with that reference in Fig. 9. We could thus calculate cross-wind integrated footprint functions as a function of actual distance by scaling the general shape according to the half-widths $dx_{0.5}$ that were calculated as per Eq. 3. Thereby, we obtained a relatively robust 1-D footprint for each derived flux, in particular for the full $F_{CWT}$ time series. Errors in the obtained dimensions were nominally subject only to uncertainties in $w^*$ and $h$, whereas uncertainties in friction velocity and lateral wind fluctuations did not need to be considered. The footprint locations were assumed simply upwind according to the concurrently measured wind direction.

Figure 10 illustrates the locations and dimensions of our 1-D footprint estimates for the two low-level (~315 m AGL) legs of RF13 (the 1$^{st}$ and the 3$^{rd}$ leg in Figs. 6-7), by overlaying the interquartile ranges (IQR) of the footprint functions as stripes on a satellite map. The IQR stripes were semi-transparently colored by the magnitude of respectively derived fluxes $F_{CWT}$. To give an idea of scales, the means and standard deviations for $dx_{0.5}$ were $1.4 \pm 0.3$ km. IQR were $3.1 \pm 0.6$ km long, ranging out to $P_{75\%}$ distances of $4.1 \pm 0.8$ km. Note, however, that substantial flux contributions are also expected much closer, namely 25% of the total from ranges between ~400 and 1000 m (cf. Fig. 9). In Fig. 10, those close-in ranges are located between the close end of the stripes and the flight path. The two flux 'hotspots' observed during the 3$^{rd}$ leg (cf. Figs. 6-7) are also conspicuous in Fig. 10, northwest of the SGP ground site.

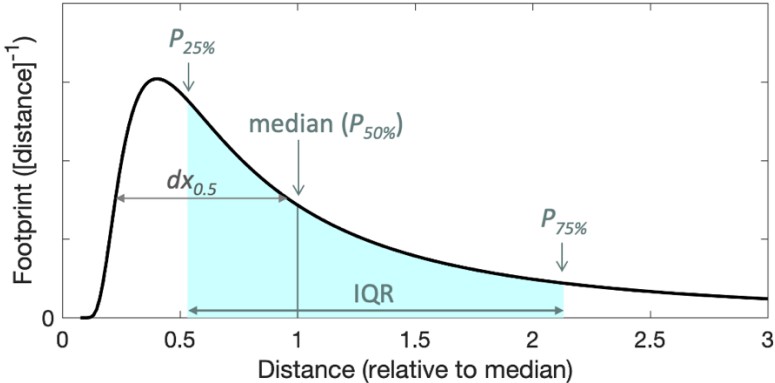

**Figure 9: Cross-wind integrated flux footprint obtained from the footprint parametrization described in Kljun et al. (2015) as a function of horizontal upwind distance from the measurement location and referenced to the footprint distribution's median ($P_{50\%}$). Marked in blue is the interquartile range, IQR, spanning from the 25th to the 75th percentiles ($P_{25\%}$ and $P_{75\%}$), and which we used for illustrating flux footprint locations and dimensions in subsequent georeferenced figures. The half-width of the function, $dx_{0.5}$, is pointed out as well.**

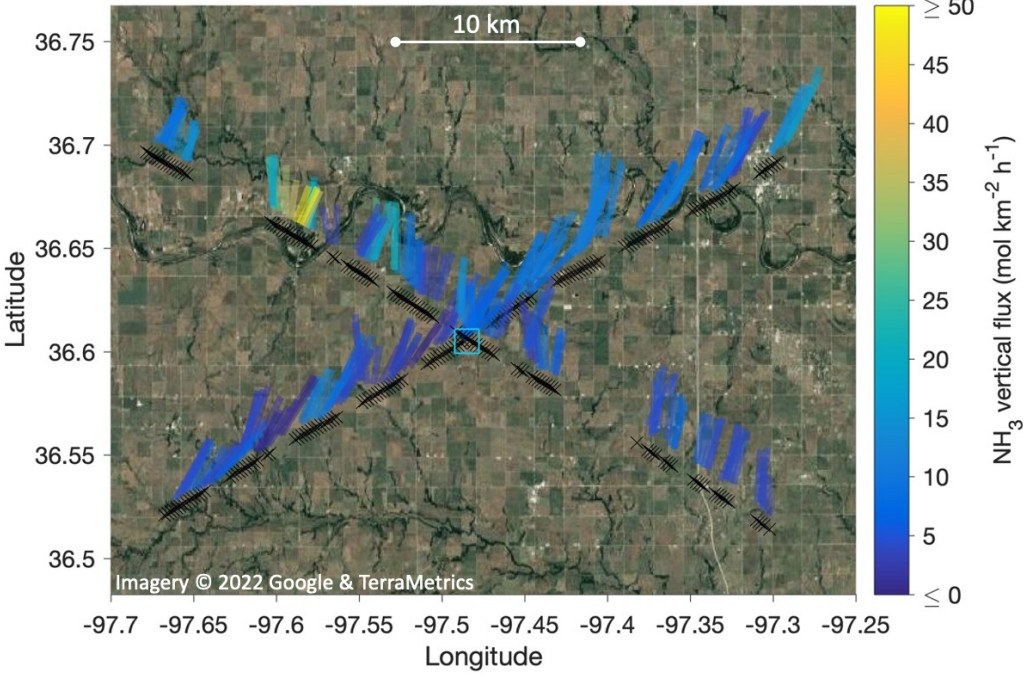


**Figure 10: Map of the flux footprint estimates for the two low-level legs of RF13, flown at 310-320 m above the SGP ground site elevation (location marked by blue square). Black crosses mark flux measurement (= aircraft) locations at 0.5-Hz resolution, converting to a spacing of 180-195 m. For each cross, a colored stripe marks the location of the interquartile range (IQR) of the respective flux footprint distribution (cf. Fig. 9), assuming locations directly upwind. I.e., winds were northeasterly. Stripe widths**
**were chosen for visibility, stripe colors correspond to the magnitude of the measured EC fluxes ($F_{CWT}$).**

Next, we compared our derived $F_{CWT}$ to the NH$_3$ area emissions expected as per the EPA's National Emissions Inventory (NEI, 2017 data, 12 km grid size). Such direct comparison implies that the $F_{CWT}$ observed at ~300 m AGL corresponded to

the net emission (or deposition) of gaseous $NH_3$ from (to) the surface, i.e., neglecting flux divergence (see above) and potential net uptake or release of $NH_3$ by aerosol. As apparent from Fig. 10, the overflown landscape is dominated by

agricultural land; emissions projected for that land presumably dominate the NEI area emissions for $NH_3$. Figure 11a shows the NEI 2017 area emissions for the day and time of RF13, for a wider area that also encompasses the HI-SCALE research flights. Emissions hotspots in that wider area relate to concentrations of intensive farming, including animal husbandry, to the south (southwest of and east of Oklahoma City) and north (northeast of Wichita). The inventory's afternoon emissions range from 5 to 12 mol $km^{-2}$ $h^{-1}$ for the areas of our flux footprints from RF13 around the SGP site (Fig. 11b). The diel

maxima (up to 13 mol $km^{-2}$ $h^{-1}$) are reached a bit later in the afternoon (~21:00). Note that the NEI map contains some unexpectedly sharp transitions of $NH_3$ emissions along lines (Fig. 11a). These lines coincide with political boundaries rather than changes in land cover or land use, specifically the boundaries of various counties, and are therefore likely the result of county-level inconsistencies in emissions reporting or NEI compilation. Shifts in the overall level of NEI area emissions are also apparent in Fig. 11b, and they correspond to both flight tracks crossing county lines.

On average, the fluxes measured during the first leg of RF13 broadly corresponded to the emissions indicated by the NEI (Fig. 11b, left). Measured $NH_3$ fluxes, slightly averaged to 0.5 Hz, corresponding to spatial scales of ~200 m, reached up to 18 mol $km^{-2}$ $h^{-1}$ but also down to –4 mol $km^{-2}$ $h^{-1}$ (i.e., net deposition). Note, however, that averaging to scales of at least 1-2.5 km was necessary to reduce random flux errors to ~100% or less. These more robust averages ranged from 1 to 11 mol $km^{-2}$ $h^{-1}$. These results provide only a 'snapshot' of the $NH_3$ emissions in the area but illustrate at least their spatiotemporal

variability in the real world. And as expected from the EC analysis, they do so at a much finer resolution than the NEI's 12-km grid. An overall similar situation was observed for the other low-level leg (Fig. 11b, right), except for the pair of peaks between 19:47 and 19:49 with emission fluxes clearly elevated above background. Figure 12a is a zoomed version of Fig. 10, focusing on the respective geographical area. The flux time series reached up to ~30 and ~50 mol $km^{-2}$ $h^{-1}$, respectively, for the two peaks, which coincided with the crossings of two $NH_3$ plumes, apparent as sharp peaks in the mixing ratio time

series (Figs. 7, 12a). Note that the first of the two peaks in the flux time series is missing its maximum due to a close-by zero measurement. The corresponding gap in the $NH_3$ mixing ratio time series was < 500 m from the plume (hereafter 'plume 1'). To avoid artifacts near data gaps (see above), the resulting gaps in the flux time series were widened, thereby engulfing the peak maximum. Noteworthy are also the widths of the respective peaks. The full width of plume 1 was only ~200 m (half-width 100 m), whereas the peak in $F_{CWT}$ had a full width of 2-3 km. Likewise, the wider plume 2 (full/half-widths of

~800/400 m) also left a $F_{CWT}$ peak ~2-3 km wide. The half-widths of the $F_{CWT}$ peaks were more poorly defined but likely ~1-2 km. These observations suggest that our application of the CWT technique to derive EC fluxes, specifically from $NH_3$ measurements at 315 m AGL, yielded a flux time series able to resolve $NH_3$ emissions at a spatial resolution of ~1-2 km along the flight track. This heuristic finding broadly agrees with the results of our error analysis that suggested that averaging to scales of ~1 km or more was typically necessary to reduce random errors to <100% (Sect. 3.8). Note that these errors

increased markedly in the vicinity of relatively localized peaks or dips in the $F_{CWT}$ time series (Fig. 11b; plumes after 19:47, also ~18:21-18:22), further cautioning against relying on $F_{CWT}$ at too short time (spatial) scales.

Choosing a smaller wavenumber for the Morlet mother wavelet (or other wavelets) can improve the localization of the $F_{CWT}$ peaks slightly. Both the standard choice of the Morlet wavelet with wavenumber 6 and the Paul wavelet of order 6 generally led to the best agreement with fluxes obtained through the ensemble-average method, and either one therefore appeared to be the best choice overall. The Paul wavelet improved localization but increased locational noise. Once averaging to >1 km, however, these differences would largely disappear. We leave it up to future studies, for instance with a more copious dataset, to explore the benefits of different choices for the mother wavelet in more detail.

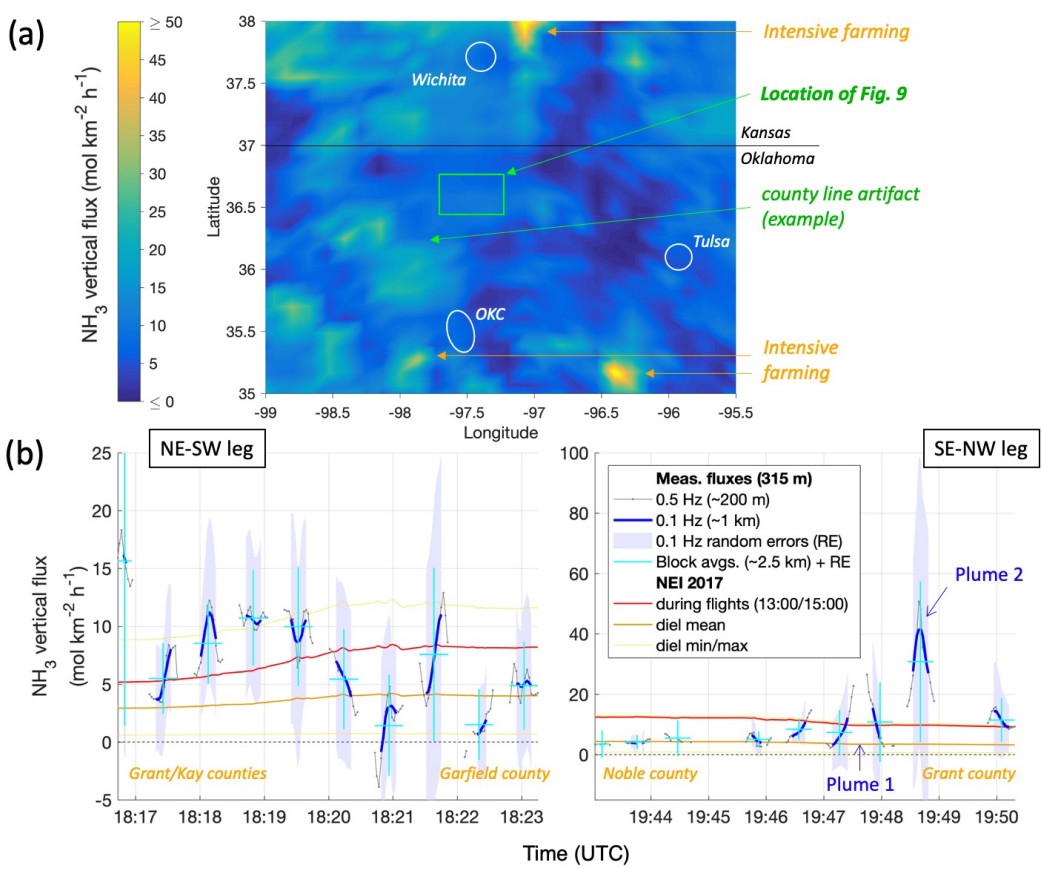

Figure 11: (a) Map of the NEI for area emissions of NH₃ for May 14 at 19:00 UTC (14:00 local time), using the same color scale as in Fig. 10. The geographic location of Fig. 10 is highlighted in green, and an example of artifacts caused by county lines (details in text) is pointed out. Major towns are indicated in white, for reference (OKC = Oklahoma City). (b) Times series of quality-controlled measurement-derived fluxes ($F_{CWT}$) during the low-level legs of RF13 (gray and blues), compared to the emissions in the NEI at the locations of the IQR of the respective footprints (as shown in Fig. 10). NEI values for 18:00 (left, 13:00 local time) and 20:00 (right, 15:00 local time) are shown in red, diel mean values in brown, the diel minimum and maximum in yellow. Measured fluxes at 0.5 Hz (~200 m spatial resolution), as used for Fig. 10, are shown in gray; moving averages to 0.1-Hz (~1 km) are shown in blue, with shades representing their random errors (RE_wave). Cyan crosses indicate block-wise (mostly ~2.5 km wide) flux averages and respective RE_wave. The times of transecting Plume 1 and Plume 2 (see text and Fig. 12) are marked by arrows.

As for plume 1, we are confident that its source was a cattle farm that the G-1 passed about 1 km downwind (Fig. 12). Evaluation of historical satellite imagery, using Google Earth, indicated that most of this specific farm's feedlots were

created in 2007/2008 and smaller expansions implemented between 2012 and 2015. However, it is not included as a source of $NH_3$ in the NEI (2017 data) point inventory, and neither resolved in the area inventory. For estimating the $NH_3$ source rate from the farm, we first used the same approach as for the fertilizer plant plume transects (Eq. 1), even though mixing throughout the boundary layer might not have occurred in this case, due to the source's proximity, so the obtained $0.6 \pm 0.1$

kg/h were likely a high estimate. As described above, the corresponding $F_{CWT}$ peak is likely missing its maximum, while its flanks are subject to large uncertainties. But putting these issues aside at first, we demonstrate how the observed fluxes and footprint considerations could be used to construct a reasonable low estimate for the $NH_3$ source rate. For that, we assumed the source area measured 200 m by 200 m (the observed plume width and about half of the farm's dimensions), and, more importantly, that it contributed maximally to the observed flux. To achieve maximum contribution, we assumed the source

area was located optimally near the footprint's maximum and exactly occupied the footprint in the crosswind dimension. The former assumption was not unreasonable given the farm's actual distance from the flight path vs. IQR locations (cf. Figs. 9 and 12). The latter assumption would also err on the intended side, given the low (>200 m) spatial resolution we conjectured for our $F_{CWT}$. Remaining conservative, we estimated the farm contributed 20 mol km$^{-2}$ h$^{-1}$ to the total observed $F_{CWT}$ at peak. With that, we obtained 0.17 kg/h for our low estimate for the $NH_3$ source rate. In combination, these considerations

constrained the cattle farm's momentary $NH_3$ emissions to between 0.2 and 0.6 kg/h. However, we want to remind that the low estimate, in this case, is subject to a flux measurement uncertainty of the order of 100%.

As for plume 2, we were not able to confidently identify a source. The plume's larger width suggested the source area was larger or farther afield compared to plume 1. It was observed 5 km downwind from the small town of Lamont, OK, where aerial/satellite (Google & Maxar Technologies; dated July 2015) and street-level imagery (Google; dated May 2013; e.g., at

36.6949° N, 97.5568° W, viewing SW) revealed storage facilities for agricultural supplies, including tanks and tank trailers with "ammonia" labels. We hypothesized that leakage associated with such storage facilities contributed to plume 2. Via Eq. 1, we obtained a $NH_3$ source rate of $2.7 \pm 0.4$ kg/h.

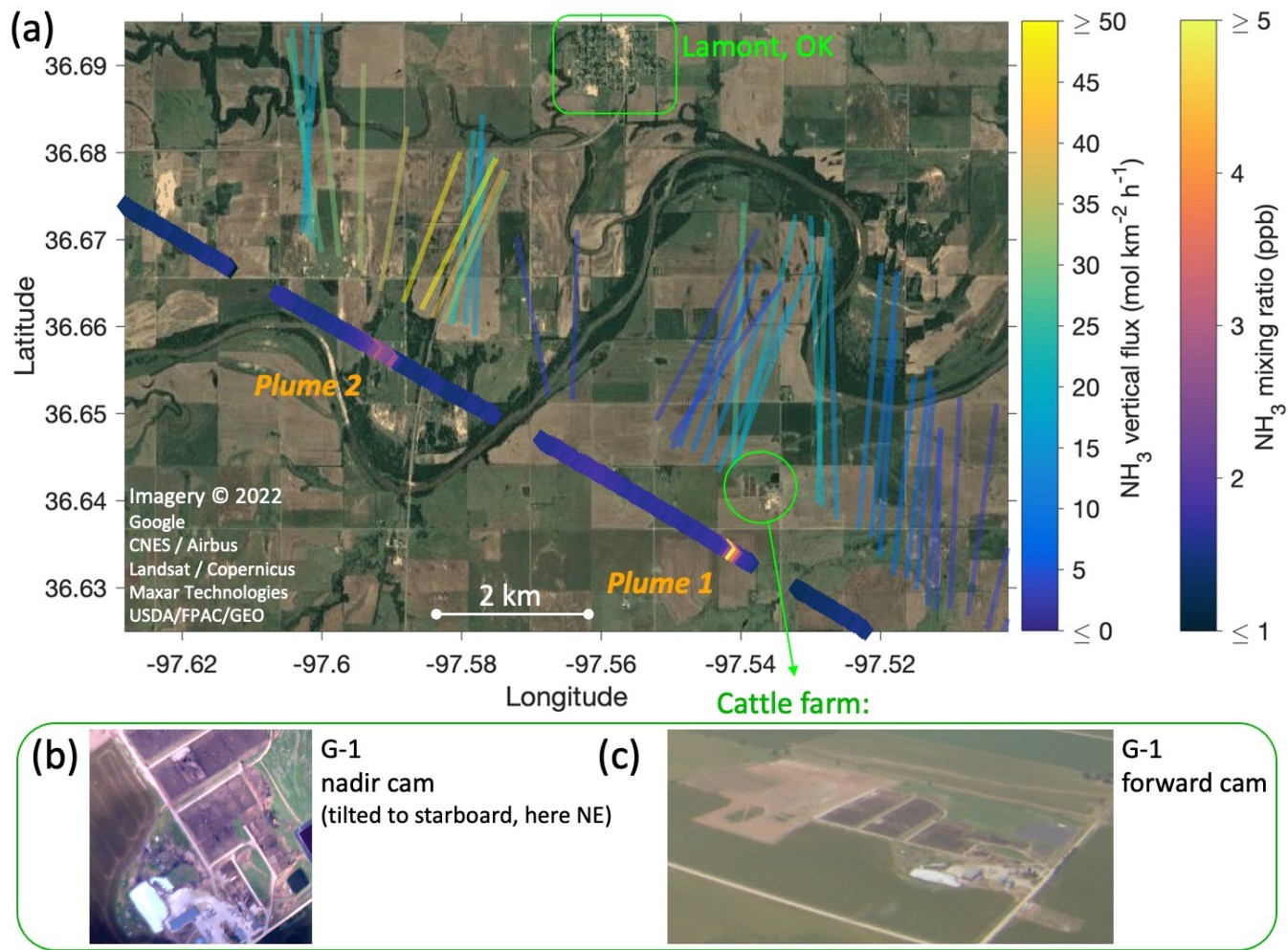

**Figure 12: (a)** close-up of Fig. 10 for the two NH$_3$ plume transects south of the town of Lamont, OK. As in Fig. 10, the IQR of the flux footprints are overlaid on satellite imagery, color-coded for the measured fluxes using the same color scale. Additionally, measured NH$_3$ mixing ratios are shown by separately color-coded markers along the flight track. **(b)** and **(c)** are photographs of the cattle farm identified as the source of plume 1, made by the nadir and forward cameras on the G-1 aircraft, respectively. (The nadir camera was slightly tilted to starboard, allowing the farm to come into view.)

## 4 Summary & Conclusions

We have presented a new mass spectrometry-based technique for detecting and quantifying NH$_3$ mixing ratios, specifically via chemical ionization using benzene cations. The technique was adapted to a CIMS that had been modified for airborne measurements, in particular an efficient sampling setup, which resulted in a highly sensitive and responsive device for measuring ambient NH$_3$. We demonstrated its capabilities by presenting results from its deployment on a G-1 aircraft during the HI-SCALE field campaign. The focus was on analyzing plume transects, and on eddy covariance (EC) analysis to derive vertical fluxes that we connected with agricultural NH$_3$ emissions through footprint considerations.

Calibration experiments revealed a humidity dependence of the sensitivity of the $NH_3.C_6D_6^+$ ion counts to $NH_3$ mixing ratios, in particular a substantial drop in sensitivity when sampling relatively dry air (RH < 20% at room temperature). In the atmospheric boundary layer, evaporation from the surface typically causes a substantial upward flux of water (often expressed as a latent heat flux). That flux would generally cause a positive bias in derived $NH_3$ fluxes if the sensitivity's humidity dependence was not considered, and a precise understanding of that dependence becomes even more important. Consequently, when we assessed that understanding rather conservatively, it became the largest source of systematic uncertainties (Sect. 3.8). We were not able to determine the mechanism behind that humidity dependence, and it remained unclear if the source of humidity in the IMR matters. In any case, we suggest careful calibrations prior to future field deployments. Active humidification of the IMR could be considered, while keeping in mind the risk of possibly introducing contaminants. Addition of a suitable dopant might also be effective in reducing the humidity dependence.

The time response of our setup to changes in $NH_3$ mixing ratios was on the order of a second. As we demonstrated, such quick response makes the instrument very well-suited for precise measurements in airborne applications and EC analysis. There was a non-negligible background signal (up to 10% of the total) that responded more slowly, on the order of a few minutes, which may become an issue in the form of high relative background signal when quickly transitioning from generally high to relatively much lower $NH_3$ mixing ratios. Corrections for that background response time could be considered, e.g., analogous to the time response correction method discussed in Nguyen et al. (2015). For this study, we did not apply such corrections but assessed them to amount to <10% of over- and <25% of underestimation for the worst-case plume transect and climb into a cleaner free troposphere, respectively. It appeared that at least the slowly responding background was due to repartitioning of $NH_3$ from walls of both the inlet line and the IMR, suggesting that attention should be paid to the designs of both the sampling setup and the IMR geometry in view of limiting wall interactions to achieve optimal performance. Also, accuracy would be improved if instrument backgrounds were determined by overflowing the full inlet instead of only the IMR; again, especially relevant for transitions to relatively lower mixing ratios.

A different, practical concern may be the toxicity of benzene, which was consumed in substantial quantities in order to achieve mixing ratios of >100 ppm (at 100 mbar) in the IMR. To avoid exposure, the instrument exhaust was routed outside of the aircraft cabin, or into a fume hood exhaust when in the lab. The risk of spillage remained, especially when refilling the benzene reservoir in field settings. One could attempt to substitute with toluene, which has seen use in lieu of benzene for some applications for that very reason (Alton and Browne, 2020).

For our EC analysis, we largely followed the example set by W2018 and used the CWT method to obtain time series of turbulent $NH_3$ fluxes. It allowed us to constrain the net atmosphere-surface exchange (here, mostly emissions) of $NH_3$ upwind of the flight path, and to do so at a higher spatial resolution than what is generally achieved via the more traditional EA method. The CIMS data featured frequent gaps due to fast zero measurements (every 42 s for 6 s), which caused comparatively larger gaps (15 s) in the flux time series. However, frequent 'zeros' are required for accurately and precisely quantifying mixing ratios, especially when experiencing fast changes, which are typical in airborne applications (cf., Lee et al., 2018), and errors in mixing ratios directly propagate into the derived fluxes. The resulting trade-off between accuracy

and spatial coverage of the flux data requires consideration. It has also become clear that longer legs would be beneficial, as many of the flux data ended up flagged and discarded due to possible edge effects of unquantified larger-scale covariance ('COI', Fig. 7). It may be useful to estimate that possible error and continue using many of these flagged fluxes, furnished with appropriate uncertainties, rather than to just discard them. Longer legs, as well as lower altitudes, are also expected to reduce systematic and random errors related to (low-frequency) turbulence (Sect. 3.8). Random errors were clearly the dominant source of uncertainty for the obtained flux time series. They often dropped below 100% only after averaging to at least ~1 km, thereby imposing effective limits to spatially resolving fluxes. We estimated that at least about half of the random error was due to instrumental noise, which could be reduced by flying at lower airspeeds.

When connecting aircraft-measured EC fluxes to emissions from (or deposition to) the ground, also vertical flux divergence needs to be considered, as discussed. Flight profiles should be planned to allow the collected data to constrain the terms in Eq. 2 as needed (e.g., Karl et al., 2013; W2018). For estimating the surface areas affecting the measured fluxes, the flux footprints must be estimated, which in general are functions of upwind distance and direction, preferably perpendicular to the flight track. Again, suitable flight planning could enable valuable additional constraints, for example via parallel legs that lead to partial footprint overlaps. For the footprint estimates in this study, we used a convenient mix of established parametrizations to obtain 1-D flux footprint functions. Required inputs, in addition to wind data, were boundary-layer depth and near-surface sensible heat flux. Estimating the latter from airborne data introduced most uncertainty regarding footprint dimensions. Alternatively, 2-D footprints could be calculated, e.g. using the full footprint parametrization proposed by Kljun et al. (2015), for which those fluctuations should lead to a lateral broadening. However, additional input parameters are then required, and the processing of the sum of obtained footprints for each flight leg will be more involved. The 2-D approach was pursued, for example, by Hannun et al. (2020) to attribute airborne CWT fluxes of greenhouse gases to different land classes.

This paper's main goal is to introduce (airborne) benzene-CIMS as a method of measuring $NH_3$ mixing ratios and fluxes, but there are also some scientific takeaways. First, we provided an overview of the variability of $NH_3$ in the lower troposphere, likely representative at least for rural Oklahoma in May (Fig. S3). Boundary layer mixing ratios spanned over one order of magnitude (broadly from 1 to 10 ppbv), but within each flight appeared vertically well mixed. Tens of ppbv were observed in plumes from a large fertilizer plant. Free tropospheric $NH_3$ mixing ratios were a factor of 3 to 10 lower and reached down to 100 pptv. A better understanding of the vertical distribution and transfer of $NH_3$ may be desirable, as a substantial fraction of $NH_4NO_3$ may actually occur in the cooler, upper layers of the atmosphere (e.g., Fig. 6b), including the free troposphere (Paulot et al., 2016; Höpfner et al., 2019). Airborne in-situ measurements as we present here could provide observational constraints, in particular if encompassing more than occasional stages of research flights and climbing sufficiently high. Appropriately planned, comprehensive $NH_3$ vertical profile measurements could also be used to improve satellite retrievals (Van Damme et al., 2015). Second, our analysis of $NH_3$ plumes and fluxes, which we tied to surface emissions, provided quantified snapshots of agriculture-related area and point sources of $NH_3$. The results were overall consistent with the NEI

inventories, but substantial point sources also seemed to be missing in the NEI. That finding is in line with recent literature arriving at similar conclusions (see Introduction).

We suggest that setups to measure EC fluxes of $NH_3$ could play an important role in providing top-down observational constraints on $NH_3$ emissions, in particular from agricultural sources. Airborne measurements in particular could help by providing regional coverage across ranges of surface properties and ecosystems. As they can resolve air-surface exchange at high spatial resolution, they could be used in conjunction with detailed information regarding concurrent agricultural practices (e.g., timing and type of fertilizer application or manure management). Ground-based deployments, on the other

hand, would more easily provide longer-term and more continuous information to cover a wider range of environmental conditions and surface activities.

An additional strength of the (TOF-)CIMS method in particular is that a range of other compounds can be quantified independently and at the same time. For benzene-CI, the obvious candidates are terpenes and dimethyl sulfide (Lavi et al., 2018), as well as a range of many other volatile organics, such as polyaromatic hydrocarbons. We have seen that switching

back and forth to another reagent ion (or several other reagent ions) works at least for iodide-CI here, multiplying the detectable range of compounds. From the $NH_3$ point of view, interesting compounds detectable by iodide-CI could be $HNO_3$, HONO, and other oxidized forms of N (e.g., Lee et al., 2018). Our experience with such a mode of operation was mixed for an airborne deployment, but it is likely more viable during ground-based deployments.

If the focus is on $NH_3$, on the other hand, it may be feasible to use benzene-CI also in a smaller, lighter and cheaper mass

spectrometer, for example using a residual gas analyzer. During HI-SCALE, $NH_3.C_6D_6^+$ was by far the dominant composition detected at $m/z$ 101, and one could have gone without the high resolution provided by an expensive TOF, as long as sensitivity was preserved.

**Data Availability**

Data from the HI-SCALE campaign is available at https://iop.archive.arm.gov/arm-iop/2016/sgp/hiscale/ (last access 24

August 2022). National Emissions Inventory 2017 data is available from the US Environmental Protection Agency, e.g. at https://www.epa.gov/air-emissions-inventories/ (last access 24 August 2022). All data used in this study are also available from the authors upon request. We used the Google Maps web mapping platform to access satellite and street-level imagery (https://www.google.com/maps; last access 24 August 2022, including used imagery).

**Author Contributions**

JF led the HI-SCALE field campaign, in particular the G-1 research flights, and prepared NEI area emissions inventory data. JAT perceived and supervised the deployment of the CIMS using dual reagent ions on the G-1. The CIMS setup and installation were realized by SS and tested and deployed by SS, ELD and BHL. DMB and JES helped with its operation, in

particular during research flights. Calibration experiments were performed and analyzed by SS, ELD and LV. SS analyzed the HI-SCALE CIMS data. Eddy-covariance analysis was performed by SS, with assistance from QP and MP. AMS data was provided by JES. MS performed the WRF-Chem model runs and provided respective details and figures. SS wrote the manuscript. All co-authors participated in paper-related discussions and commented the manuscript.

### Acknowledgments

We thank the ARM Aerial Facility team, and everyone involved with the HI-SCALE campaign for the outstanding support and collaboration. Particular thanks go to Payload Director John Hubbe whose support from the planning phase onwards assured the seamless integration of the CIMS into the G-1. We also appreciate the help of Dennis Canuelle in the machine shop, of Lexie Goldberger during test flights, and valuable discussions regarding the benzene-CI technique with Tim Bertram and Gordon Novak (University of Wisconsin-Madison).

### Financial Support

SS was supported for this research by the European Commission through the Marie Skłodowska-Curie Actions (OXFLUX Global Fellowship, project 701958) and by the Academy of Finland (Postdoctoral Researcher grant no. 310682, and grant no. 337550 under the Flagship programme). The UW CIMS deployment and data processing were also funded by grants from the U.S. Department of Energy (DOE) Office of Science (DE-SC0021097, DE-SC0011791) and Pacific Northwest National Laboratory (contracts 243766 T.O. 276416). ELD was supported by the National Science Foundation Graduate Research Fellowship under grant no. DGE-1256082. JS and JF are supported by the DOE Atmospheric System Research (ASR) program. MS was supported by the DOE Office of Science, Office of Biological and Environmental Research through the Early Career Research Program. Pacific Northwest National Laboratory (PNNL) is operated for DOE by Battelle Memorial Institute under contract DE-AC05-76RL01830. The contributions by PNNL were also funded by the DOE's Atmospheric Radiation Measurement (ARM) User Facility.

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
