# Peer review of "Airborne flux measurements of ammonia over the Southern Great Plains using chemical ionization mass spectrometry"

_Atmospheric Measurement Techniques, 2022_

## Referee Comment (RC1)

Review of "Airborne flux measurements of ammonia over the Southern Great Plains using chemical ionization mass spectrometry," Schobesberger et al., AMT (2022)

**Summary**

This paper presents new airborne measurements of ammonia (NH3) using a benzene CIMS and shows the application of the data to eddy covariance flux calculations over agricultural land. It describes the measurement method, calibration and sampling challenges, and then provides a detailed description and some analysis of NH3 fluxes. English is generally good. The number of figures is appropriate, but some of them need tweaking for readability and comprehension.

Despite how many minor comments appear below, I think this is a good paper and suitable for publication in AMT.

**Specific Comments**

L14 – 20: The level of detail in the opening sentences of the abstract is more than needed. The first sentence would be sufficient, and the next four sentences could be removed.

L56: Critical load thresholds are also exceeded in North America (https://www.sciencedirect.com/science/article/abs/pii/S0048969719329109), but I'm not sure how much of that is attributable to NH3.

L64: This is a run-on sentence and may have some grammatical errors too.

L114: OPALS has not flown or been published. This sentence should be deleted and the following sentence adjusted.

L139 – 145: Recommend moving the sentences starting with "In this paper" and "The use of" to the next paragraph.

L229: Previous discussion has highlighted the high frequency of the TOF. Why is the sample frequency limited to 2Hz here?

L278: "would reduce the gap between the curves…" Is there a reason why you did not do this, if it would collapse the curves?

Sect. 3.1: I expected to find accuracy numbers at the end of this discussion. I know it is discussed later, but might also be appropriate to mention here.

Sect. 3.2: What is the time constant for the longer decay? Discussion mostly focuses on the first decay.

L309: Nguyen et al. (https://www.pnas.org/doi/abs/10.1073/pnas.1418702112) discuss a potential correction for this inlet hysteresis in their supplement (related to HNO3). Not suggesting that the authors implement that here, just wanted to make you aware of it.

Figure 3: I found it difficult to follow this figure and the caption. It has multiple legends and a lot of different experiments and assumptions. Not sure what the fix is. Maybe break into several panels, or replace the symbols with lines, or something else to simplify it?

L331: is this precision based purely on counting noise / background? Does this number change with humidity?

L363: If there is hysteresis as described, then ascending and descending profiles might have different shapes. It is hard to glean this from Fig. S3.

L382: It would be helpful for some readers to provide some references for this box budget technique.

Figure 5: This figure needs a little work. In Panels (a) and (b), coloring by altitude is unnecessary (and indeed confusing as the same color scale is used for a different parameter in (c)). Also, the colored text on the right hand side of (b) is hard to read. Also in (b), the arrows could be removed; a single arrow for average wind direction would suffice.

Figure 6: wrong caption. Also, the y axis in the top panel needs a better label, panels should be lettered, and it might be worthwhile to convert the x axis to distance.

Sect. 3.6: I think it would be helpful to show the power spectra for w', NH3' and T', along with cospectra, in the main text instead of the supplement, given the concerns about spectral attenuation.

L444: flux code is available on github (https://github.com/AirChem/FluxToolbox)

L450: "variety of interval lengths." Over what range?

L454: It is better to use distance than time for the independent variable in airborne fluxes. So instead of 1 minute, 6 km?

L469: It is still beneficial to detrend the data, especially if you are including larger scales in your calculated fluxes. Detrending is also necessary to get a clean lag-covariance plot.

L478: How long is the lag time?

L529: I am not a micrometeorologist, but I have come to understand "constant flux layer" as meaning "the altitude range over which changes due to divergence are smaller than our flux precision."

L534: Need to also mention the first term (storage).

L540: Wolfe et al. 2015 (https://agupubs.onlinelibrary.wiley.com/doi/full/10.1002/2015GL065839) show non-linear divergence for species with strong T-dependence, like PAN and HPALD. Could there be similar effects for NH3 fluxes due to gas-particle partitioning?

Figure 9: Blue-on-green is not a great contrast when printing this in color. Maybe switch to a different colorbar?

Figure 10: it is very hard to see the blue arrow on top of the blue background.

L623: effective spatial resolution is also limited by inherent random error related to turbulence. Errors in 1Hz (~100m) fluxes can exceed 100%.

L624: There may also be advantages to switching to a different mother wavelet; for example, some wavelets have better localization in space (at the expense of less frequency localization) and may be better suited for large point sources. This is certainly an area worthy of further research, especially if we had a way to validate which wavelet gave the "best" answer.

Figure 11: I find the use of the same color bar for 2 different quantities confusing.

L708: I expect other sources of uncertainty dominate over any locational noise in the footprint.

L711: Hannun et al. 2020 (https://iopscience.iop.org/article/10.1088/1748-9326/ab7391) have had success with 2-D footprint modeling.

**Technical Comments**

L53: delete "its"

L136: delete "And"

L166: delete ", specifically"

L211: delete "of our CIMS"

L398: replaced "transiting" with "transecting"

L595: might be worthwhile to mark these cities on the map.

L595: "emissions range from"

L617: It might be helpful to label the numbered plumes in Fig. 10.

L618: "series were widened"

L657: delete "in this paper"

---

## Author Comment (AC2)

**Response to reviewers' comments (received Sep 30 and Oct 31, 2022) Dec 14, 2022**

Title: "Airborne flux measurements of ammonia over the Southern Great Plains using chemical ionization mass spectrometry" Paper amt-2022-244 Authors: S. Schobesberger, et al.

We thank both reviewers for their thoughtful study of our manuscript, their overall positive statements, as well as the many productive comments.

We believe the implemented changes have substantially enhanced our manuscript, while satisfying the referees' concerns.

A major change was the addition of an analysis of systematic and random flux errors. These changes encompass a new additional section (Sect. 3.8), table (Table 1), additional figures (Figs. 8, S5, S8, S9), additions to Fig. 11 (formerly Fig. 10), and a few adaptations and additions to the rest of the text, in particular the remainder of Chapter 3 and the Summary & Conclusions (Chapter 4).

Our point-by-point replies are given below (blue Times New Roman font) following each of the reviewers' comments, which are repeated in full (black Arial font). Reproduced text from the revised manuscript is set in black and green bold Calibri font, green marking changes or additions.

Important updates of or additions to figures, as well as the new Sect. 3.8 are appended following our replies (using the black Times New Roman fonts from the journal's template).

**\*\*\*\*\**

**Referee #1 (Glenn Wolfe):**

**Specific Comments**

L14 - 20: The level of detail in the opening sentences of the abstract is more than needed. The first sentence would be sufficient, and the next four sentences could be removed.

**Good suggestion, abstract shortened accordingly.**

L56: Critical load thresholds are also exceeded in North America (https://www.sciencedirect.com/science/article/abs/pii/S0048969719329109), but I'm not sure how much of that is attributable to NH3.

**Thanks for pointing out that work. We extended the sentence: [...] and contributes to critical reactive N load exceedances in North America (Walker et al., 2019).**

L64: This is a run-on sentence and may have some grammatical errors too.

Thanks for pointing out. Added a missing word and also shortened the sentence.

L114: OPALS has not flown or been published. This sentence should be deleted and the following sentence adjusted.

**Deleted, and erstwhile following sentence changed to**

Airborne deployments have so far favored closed-path systems, as introduced above.

L139 – 145: Recommend moving the sentences starting with "In this paper" and "The use of" to the next paragraph.

Yes, it probably flows better that way. Done.

L229: Previous discussion has highlighted the high frequency of the TOF. Why is the sample frequency limited to 2Hz here?

We acquired data at 2 Hz as a compromise. We indeed expect that data could have been acquired at for instance 10 Hz just as readily. The main advantage of acquiring at 2 Hz, vs 10 Hz, is of course that 2-Hz data requires considerably less storage space and is faster to handle than 10-Hz data. Using the CIMS for eddy covariance analysis had not been considered a main goal of the deployment for HI-SCALE, while we estimated that a time resolution of 2 Hz would be acceptable for deriving fluxes if we would want to. Indeed, the co-spectra (Figs. S4, S5) show quickly diminishing flux contributions as frequency increases towards 1 Hz. The corresponding ogives (not shown) indicate that 90% of co-spectral power occurs at frequencies < 0.14-0.18 Hz; or 92% to 96% at < 0.2 Hz, which is a fifth of our Nyquist frequency of 1 Hz. With that, we crudely and conservatively estimate that quintupling our sampling rate to 10 Hz would add at most 4% to 8% of flux.

We added the following sentence:

A data acquisition rate of 2 Hz, rather than e.g. 10 Hz, was a compromise that reduced requirements for data storage and computing times during data processing, while potentially induced errors in obtained fluxes were deemed acceptable (see Sect. 3.8 for details).

L278: "would reduce the gap between the curves..." Is there a reason why you did not do this, if it would collapse the curves?

It remains unresolved, if sensitivity depends on the humidity of the sample or on the humidity in the IMR overall. Previous studies (e.g., Lee et al., 2014) have mostly considered the latter, so we chose to do the same here. The better agreement between the curves when plotting vs. sample humidity may be a coincidence, as the respective calibration setups differed in multiple aspects, including IMR geometry. Therefore, we want to leave the question open and plot vs. humidity in the IMR, as done previously. We added the following side sentence for clarification:

As the CIMS devices (blue vs. orange) differed in particular in their IMR geometries, we leave it up to future studies to ascertain if sensitivity to NH3 relates to the humidity of only the sample (absolute or relative) or the humidity in the IMR.

Sect. 3.1: I expected to find accuracy numbers at the end of this discussion. I know it is discussed later, but might also be appropriate to mention here.

Section 3.3 is devoted to "quantification", and we believe it is more appropriate and better-flowing to discuss there, how the calibration results were used for obtaining mixing ratios. However, we added a sentence there (Sect. 3.3) to explicitly point out how uncertainties in sensitivities lead to systematic error in the obtained mixing ratios:

We could not rigorously establish the uncertainty in used sensitivities (see also Sect. 3.8), but a conservative estimate of ±1 ncps/pptv would result in a systematic error for the reported mixing ratios of typically 20% to 30%. Figure 4 illustrates the process of obtaining NH3 mixing ratios for a segment of RF6 [...]

Sect. 3.2: What is the time constant for the longer decay? Discussion mostly focuses on the first decay.

It varied, and we only performed three 'zeros' at the sample inlet tip that were long enough to allow for determining the time constant for the longer decay to background levels. Based on these, that time constant was  $4 \pm 2$  min. Added that info at the end of the 1st paragraph in Sect. 3.2:

That background level is responding more slowly. For three long 'zeros' at the sampling inlet tip, that slower decay followed a time constant of  $4 \pm 2$  min.

L309: Nguyen et al. (https://www.pnas.org/doi/abs/10.1073/pnas.1418702112) discuss a potential correction for this inlet hysteresis in their supplement (related to HNO3). Not suggesting that the authors implement that here, just wanted to make you aware of it

Thanks, this is a salient point that could probably also be applied to the background "base level" hysteresis we experienced, e.g. to improve the accuracy of boundary layer-free troposphere transitions. We added a reference to that study in the Conclusions:

[...] which may become an issue in the form of high relative background signal when quickly transitioning from generally high to relatively much lower NH3 mixing ratios. Corrections for that background response time could be considered, e.g., analogous to the time response correction method discussed in Nguyen et al. (2015). For this study, we did not apply such corrections but assessed them to amount to <10% of over- and <25% of underestimation [...]

Figure 3: I found it difficult to follow this figure and the caption. It has multiple legends and a lot of different experiments and assumptions. Not sure what the fix is. Maybe break into several panels, or replace the symbols with lines, or something else to simplify it?

We tried to unload Fig. 3 by splitting it into two panels, use of lines, restyling the legends, and slightly restructuring the caption.

L331: is this precision based purely on counting noise / background? Does this number change with humidity?

Yes, it is based purely on background counts. We added now the word "pure" to better clarify. In two of those four 'zero' experiments, we tested turning on IMR humidification (mentioned also a few lines above), upon which noise increased by a factor of 2 or 4. With the then expected humidity (0.35 mbar), sensitivity should have increased by a factor of at least 6 (cf. Fig. 2), and the resulting precision in terms of mixing ratios (pptv) would be 5 to 10 pptv, or about a factor of two better than before humidifying. We would, however, have expected precision to be the same, or possibly worse (for instance, if contaminant NH3 was introduced via the humidification). It appears likely that humidifying the IMR was less 'potent' in raising sensitivity than humidifying the sample, as speculated in Section 3.1.

Anyway, we decided not to include these results in the manuscript to avoid possibly adding more confusion than necessary. Besides, they are based on only two tests, and they ultimately result in more speculation.

L363: If there is hysteresis as described, then ascending and descending profiles might have different shapes. It is hard to glean this from Fig. S3.

We agree, but no such hysteresis was observed.

Two flights contained both ascending and descending profiles out of and into the boundary layer: RF6 and RF8. Only the profiles from RF8 displayed an *apparent* hysteresis. It was, however, the 'wrong way round': there was a sharp drop of  $[NH_3]$  when climbing through 800 m, complete at 1270 m, followed 30 min later by a gentle rise of  $[NH_3]$  during descent, starting already at ~1500 m until descending through 800 m.

Very likely, the hysteresis effects (due to partially slowly responding instrument backgrounds) were considerably smaller than natural variability during our transitions into and out of the free troposphere.

L382: It would be helpful for some readers to provide some references for this box budget technique.

**Added the following sentence:**

For previous instances of that simple mass balance approach, see, e.g., Turnbull et al. (2011) and references therein.

Figure 5: This figure needs a little work. In Panels (a) and (b), coloring by altitude is unnecessary (and indeed confusing as the same color scale is used for a different parameter in (c)). Also, the colored text on the right hand side of (b) is hard to read. Also in (b), the arrows could be removed; a single arrow for average wind direction would suffice.

We simplified content and colors in Fig. 5, panel (a) and (b), making it clearer overall.

Figure 6: wrong caption. Also, the y axis in the top panel needs a better label, panels should be lettered, and it might be worthwhile to convert the x axis to distance.

Apologies for the mistake; we inserted to correct caption. And further updated the figure as requested.

Sect. 3.6: I think it would be helpful to show the power spectra for w', NH3' and T', along with cospectra, in the main text instead of the supplement, given the concerns about spectral attenuation.

As spectral attenuation ultimately turned out not to be a large concern (see also new Sect. 3.8, discussing errors), we prefer to leave the power and co-spectra in the supplement. Note that the CWT co-spectra are in principle also shown in Fig. 7.

L444: flux code is available on github (https://github.com/AirChem/FluxToolbox)

Thank you for pointing that out! That current version on GitHub indeed includes some additional helpful scripts we had not been aware of. We updated the manuscript to include the reference to the code on GitHub.

L450: "variety of interval lengths." Over what range?

**Including that information now as follows:**

We calculated *FEA* for a variety of interval lengths (from 1 to 10 min), for assessing the general feasibility of the EA method for obtaining EC fluxes from our dataset.

L454: It is better to use distance than time for the independent variable in airborne fluxes. So instead of 1 minute, 6 km?

**Indeed:**

The results are internally consistent, as  $F_{EA}$  using shorter time intervals (down to 1 min, corresponding to ~6 km, cyan) broadly average to the results using longer intervals.

L469: It is still beneficial to detrend the data, especially if you are including larger scales in your calculated fluxes. Detrending is also necessary to get a clean lag-covariance plot.

Thanks for pointing that out. Lag times were only determined here via the EA method, where stationarity was required. And large-scale trends were anyway not an issue in our case. With that, and to enhance

readability, we prefer to refrain from a discussion of the potential benefits of detrending. But we added the word "generally" to somewhat weaken our renunciation of detrending. That said, new Fig. S5 does now present also wavelet-calculated lag-covariance functions.

L478: How long is the lag time?

The lag time for RF13 was actually ~zero. (The mean of 11 independent periods was  $-0.01 \pm 0.22$  s; see also Fig. S4 ... now also with corrected x-axis for the lag covariance plots). Added that information to the caption of Fig. S4 and added the following to the main text:

We applied lag time as obtained by the EA flux calculations (typically < 1 s; zero lag was used for RF13; see Fig. S4). See also reply to referee #2 comment 9 for more detail on lag times.

L529: I am not a micrometeorologist, but I have come to understand "constant flux layer" as meaning "the altitude range over which changes due to divergence are smaller than our flux precision."

Our understanding is based on the description in the American Meteorological Society's online glossary (https://glossary.ametsoc.org/wiki/Constant\_flux\_layer; last access 29 Nov 2022). It refers to the altitude range over which relative changes of flux are within 10%, i.e., not based on measurement performance.

L534: Need to also mention the first term (storage).

**Added:**

[...] where the first term on the right side is storage, and the second horizontal advection [...]

L540: Wolfe et al. 2015 (https://agupubs.onlinelibrary.wiley.com/doi/full/10.1002/2015GL065839) show nonlinear divergence for species with strong T-dependence, like PAN and HPALD. Could there be similar effects for NH3 fluxes due to gas-particle partitioning?

Good point. NH3 is expected to (and we indeed observed it to) partition more efficiently into the particle phase with colder temperatures and higher humidity, i.e., within a turbulently mixed layer, increasingly with increasing height.

Included now  $pNH_4$  into the respective sentence. (It had been missing, as more precise pNH4 results had become only available during drafting of the manuscript.)

Figure 6 also indicates that particulate NO3 and *p*NH4 concentrations within the boundary layer tended to increase with altitude [...]

More importantly, we added a mention of potential impacts of T-mediated gas-particle partitioning to flux divergence. As our dataset was too sparse for a careful analysis of flux divergence (as described), we leave it at that.

Once could expect such linearity also for NH3, as its boundary-layer lifetime against oxidation is weeks to months (Diau et al., 1990), while its gas-particle partitioning can be assumed to be in equilibrium, at least outside plumes. Flux divergence may then be obtained more directly by measuring fluxes at multiple altitudes, as suggested, e.g., in W2018. For species with vertically inhomogeneous source or sink rates, however, flux divergence may be non-linear (Wolfe et al., 2015). Indeed, the partitioning of NH3 into the particle phase is expected to be enhanced at lower temperature, which generally decreases with height within the boundary layer (Sect. 3.5, Fig. 6). In any case, we were unfortunately unable to consistently investigate flux divergence for this study, [...]

Figure 9: Blue-on-green is not a great contrast when printing this in color. Maybe switch to a different colorbar?

We actually obtained very good contrast when printing, at least from the closest available printer. We tested some other color schemes too but found the current blue-to-yellow ("parula") colormap the best perceptual and colorblind-friendly choice (and also with fair-to-good contrast against the background across its range).

Figure 10: it is very hard to see the blue arrow on top of the blue background.

**Changed to green.**

L623: effective spatial resolution is also limited by inherent random error related to turbulence. Errors in 1Hz (~100m) fluxes can exceed 100%.

The manuscript now contains ample discussion of errors in the new Section 3.8. Indeed, random errors were dominant, and often exceeded 100% even down at 0.1 Hz ( $\sim$ 1 km). The part of now-Sect. 3.9 was correspondingly extended as well:

The half-widths of the  $F_{CWT}$  peaks were more poorly defined but likely ~1-2 km. These observations suggest that our application of the CWT technique to derive EC fluxes, specifically from NH3 measurements at 315 m AGL, yielded a flux time series able to resolve NH3 emissions at a spatial resolution of ~1-2 km along the flight track. This heuristic finding broadly agrees with the results of our error analysis that suggested that averaging to scales of ~1 km or more was typically necessary to reduce random errors to <100% (Sect. 3.8). Note that these errors increased markedly in the vicinity of relatively localized peaks or dips in the  $F_{CWT}$  time series (Fig. 11b; plumes after 19:47, also ~18:21-18:22), further cautioning against relying on  $F_{CWT}$  at too short time (spatial) scales.

L624: There may also be advantages to switching to a different mother wavelet; for example, some wavelets have better localization in space (at the expense of less frequency localization) and may be better suited for large point sources. This is certainly an area worthy of further research, especially if we had a way to validate which wavelet gave the "best" answer.

Indeed. We did test other than Morlet mother wavelets (DOG and Paul) and various wavenumbers/orders/derivatives, in particular with the more localized point sources in mind. The DOG wavelets appeared the most localized but also to introduce lots of obvious noise. Overall, the covariances obtained either via the "standard choice" (Morlet, wavenumber 6) or via the Paul wavelet of order 6 averaged most closely to those derived directly from ensemble averages. As the Paul-6 wavelet exhibited better localization than the Morlet-6 wavelet, it may be the better choice for our application. The raw covariance time series showed higher variability, but that is smoothed out once averaging to upwards of 1 km, which is where our suggested effective spatial resolution lies, and also where random errors decrease to reasonable levels. The end result is then very similar to that obtained via the Morlet-6 wavelet. So, although Paul-6 may have been the slightly better choice, we concluded that this particular manuscript is not the best place to investigate the suitability of various wavelets and proceeded with the well-performing Morlet-6 wavelet that had already been the choice wavelet in many preceding airborne EC studies. However, with the interest now raised, we slightly extended the respective discussion in now-Sect. 3.9: Choosing a smaller wavenumber for the Morlet mother wavelet (or other wavelets) can improve the localization of the FCWT peaks slightly. Both the standard choice of the Morlet wavelet with wavenumber 6 and the Paul wavelet of order 6 generally led to the best agreement with fluxes obtained through the ensemble-average method, and either one therefore appeared to be the best choice overall. The Paul wavelet improved localization but increased locational noise. Once averaging to >1 km, however, these differences would largely disappear. We leave it up to future studies, for instance with a more copious dataset, to explore the benefits of different choices for the mother wavelet in more detail.

Figure 11: I find the use of the same color bar for 2 different quantities confusing.

**Changed to a different color scheme for the mixing ratios.**

L708: I expect other sources of uncertainty dominate over any locational noise in the footprint.

Yes. Removed that statement.

L711: Hannun et al. 2020 (https://iopscience.iop.org/article/10.1088/1748-9326/ab7391) have had success with 2-D footprint modeling.

Thanks for pointing out that interesting paper. Some of us at least have missed it. Replaced the reference to "future work" with:

The 2-D approach was pursued, for example, by Hannun et al. (2020) to attribute airborne CWT fluxes of greenhouse gases to different land classes.

**Technical Comments**

L53: delete "its" Done.

L136: delete "And" Done.

L166: delete ", specifically" Done.

L211: delete "of our CIMS" Done.

L398: replaced "transiting" with "transecting" Done.

L595: might be worthwhile to mark these cities on the map. Done.

L595: "emissions range from" Done.

L617: It might be helpful to label the numbered plumes in Fig. 10. Done.

L618: "series were widened" Done.

L657: delete "in this paper" Done.

**\*\*\*\*\**

**Referee #2 (anonymous):**

**Specific Comments**

1. Humidity dependence is a major issue that the authors seem to be aware of and they used a thoughtful approach of humidity dependent sensitivity correction based on pH2O in the IMR. However, given a typically strong flux of water vapor in the troposphere and if there is a humidity dependence, the question arises if the flux contribution may not come just from NH3 but from the high frequency fluctuation in the H2O loss? This issue is neither discussed nor quantified.

That flux contribution from the upward water flux was indeed a concern in principle. However, we did correct our NH3 measurements using high-frequency humidity measurements, and also investigated the magnitude of those corrections in terms of obtained NH3 fluxes. That magnitude was generally small (Fig. R1), and ultimately, discussion of that potential issue had not made it into the manuscript. In terms of legwide averages, neglecting humidity corrections would change the absolute NH3 fluxes by 6% to 9%. If we use the steeper humidity dependence found later (blue in Fig. 2), NH3 fluxes change, on average, by 12% to 25%, which likely approaches the upper bounds of errors to expect from this correction.

Figure R1 is included in the revised supplement (Fig. S8). We also now discuss this issue among other error sources in the new Section 3.8. The variability of associated systematic error estimates is also shown in the new Fig. S9.

**Figure R1:** Top: Fluctuations of vertical wind and NH3 mixing ratio ([NH3]), with (blue) and without (red) accounting for humidity-dependent sensitivity, and with using a steeper humidity dependence (black; cf. Fig. 2), for an excerpt of the 1st leg of RF13 (14 May). Center: Fluctuations for water partial vapor pressure, calculated from dew point measurements (leg-wide flux:  $0.078 \pm 0.037$  torr m s-1). Bottom: CWT flux obtained for NH3 with (blue) and without (red) accounting for humidity-dependent sensitivity to NH3, as well as when using the steeper humidity dependence (black).

2. A simultaneous comparison with other methods (e.g. Picarro NH3) would have been more reassuring especially given the humidity issues and unexplained sensitivity drift before and after the flights.

Definitely. Unfortunately, no other NH3 measurements were available during the field deployment. However, we believe that the uncertainties of the presented method are now discussed in sufficient depth in the revised manuscript.

3. It is surprising why the zero measurement results in only 1-2 orders of magnitude signal decay (e.g. Figure 3). As the ambient concentrations may vary several orders of magnitude, it seems therefore uncertain if the instrument will be able to resolve variabilities spanning orders of magnitude within a short amount of time like it is a case when going in and out of the plume.

There is indeed notable background NH3 signal during zero measurements, as discussed. But as we have seen (e.g., Figs. 3-5; Sect. 3.2-3.4), the background responds on a timescale of several minutes, and therefore plume transects are a relatively small problem. We estimated a worst-case bias of 10% for the fertilizer plant plume transects (Sect. 3.4). A greater potential concern are quick transitions from generally high to relatively much lower NH3 mixing ratios. The closest we came to that was during climbs into the free troposphere. The manuscript discusses that issue in the Summary & Conclusions, including estimates for worst-case biases.

4. NH3 concentrations (and fluxes due to flux divergence) in the PBL and free troposphere are expected to change with altitude. However, surprisingly the concentrations look suspiciously stable for some of the RFs (e.g., Fig 3, the flight on the left RF14 or RF15 which are plotted using a similar color shade). The lack of changes in NH3 concentrations across such a broad altitude range (1000-4200 m) looks somewhat odd and potential issues with instrumental background should be excluded.

The potential issues with instrument background, we believe are sufficiently discussed (see also previous response). Consequently, we also believe that these issues produced minor occasional biases at worst. See also our response to referee #1's comment on L363, discussing the absence of obvious hysteresis effects. To streamline the paper, we kept the discussion of the NH3 vertical profiles brief. It is found at the end of Sect. 3.3, and some additional detail is found in the Supplement ("Overall NH3 levels observed during IOP1"). We believe that the key behind broad ranges of altitudes with little changes in NH3 concentrations lies in the meteorology of those days, coupled with the situation that NH3 measurements were only conducted during parts of each flight. (Please take special note also of the logarithmic scaling for the concentrations.) In the case of RF14 (broadly constant NH3 mixing ratios between 1000 and 4200 m), it was a cool and humid day, with a dense cover of low clouds. Up to ~3500 m, almost all NH3 mixing ratios largely resulted from interactions of NH3 with cloud water, with broad uniformity in the altitude range, and effectively decoupled from potential surface emissions. The situation was very similar for RF16. Generally though, a more detailed discussion of the measurement results at large in the paper would probably distract from its main goal of presenting and showcasing the measurement technique.

5. It is unclear how the data are normalized to primary ions and/or dimers. The NH3 signal does not only depend on the ambient NH3 concentration but also on the variability in the primary ion signal (C6D6 – not temperature controlled). The lack of insight into factors behind the changes in sensitivity sounds like a missed opportunity which should not be left for other papers to investigate. The reader specifically wonders how stable primary ions were throughout research flights and if heating of the benzene reservoir could be beneficial in improving this stability.

**The normalization procedure is described in the 2nd-to-last paragraph of Sect. 2.1:**

Typical reagent ion count rates were  $2-3 \times 10^6$  cps of  $C_6D_6^+$  and  $2-4 \times 10^5$  cps of  $(C_6D_6)_2^+$ . [...] NH3 was quantified from the normalized count rate of NH3.  $C_6D_6^+$  adduct ions. Normalization was to  $10^6$  cps of  $C_6D_6^+$ , i.e., measured counts per second (cps) of NH3.  $C_6D_6^+$  would typically be divided by a factor of 2-3 to obtain normalized counts per second (ncps).

During calibration experiments (with a presumed steady supply of NH3), the NH3.C6D6+ ion counts responded more directly to changes of the C6D6+ primary ions than to those of the (C6D6)2+ dimers (or the sum; though they are also coupled of course), and normalization by  $[C_6D_6^+]$  gave the simplest result. The high-frequency (1-Hz) stability of  $[C_6D_6^+]$  during flights was ~6e3 cps, i.e., <0.3%, and therefore had practically no effect on signal from ambient NH3. We did observe occasional slower drifts, typically changes in primary ion signal of up to 13% over 1-3 min. We hypothesize that those drifts were related to temperature changes of the benzene reservoir, as they sometimes appeared to coincide with ~2-K drifts in cabin air temperature. There was, unfortunately, no temperature measurement at the benzene reservoir. An additional observation was that the primary ion signal often started off about 15% to 18% low upon switching from negative to positive polarity. It would take ~10 min to reach a stable (except as described above) value between 2e6 and 3e6 cps. We speculate that behavior is due to the re-stabilization of ion guidance elements in the atmospheric-pressure interface of the mass spectrometer, which may have been specific to our instrument or may be a general feature.

We added most of that information to the manuscript, in similar wording as this response – one sentence at the end of Sect. 2.1, the remainder and the beginning of Sect. 3.3.

6. The setup looks very neat overall, but I wonder if it has been tested for changes in ambient pressure, especially that that no calibrations or targets were performed during the flight.

Thanks, it was pretty neat package indeed! We did not specifically test for changes in ambient pressure, as the instrument was equipped with a motor-controlled variable orifice as the inlet into the IMR, and the IMR was pumped via a servo-controlled valve. Both orifice and valve were computer-controlled to maintain inlet mass flow and IMR pressure. That setup is described in Sect. 2.1, and the inlet is described in detail in Lee et al. (2018), including assessments of stabilities of inlet mass flows, pressures and calibrations during flight deployment during the WINTER campaign.

For the HI-SCALE flight deployment here, inlet flow and IMR pressures were stable within <1.7% and <1%, respectively (standard deviations at 1 Hz), i.e., their variability had negligible effects on results, in particular given other, more important error sources (see new Sect. 3.8).

7. It is great to see eddy covariance estimates for ammonia. However, the flux methodology shows great potential for improvement. The introduction does not give credit to all the progress achieved in the multiple airborne EC campaigns (e.g. CABERNET, CARAFE) that have compared FFT and wavelet fluxes. I like the IRQ footprint contribution, but it would be interesting to look at the ratio between FFT and wavelet fluxes for different RFs and shed light on flux uncertainties (see #8).

The benefits of using continuous wavelet transforms (CWT) for airborne EC is briefly discussed in Section 3.6, including some references. But true, the Introduction fell short in bringing up specific previous efforts in the field: of airborne EC campaigns more generally, and regarding progress in applying CWT more specifically. We added now the following text to the paragraph that introduced airborne EC:

[...] Consequently, airborne EC has been applied for more than 30 years (e.g., Lenschow et al., 1980; Lenschow et al., 1981; Desjardins et al., 1982; Ritter et al., 1990; Ritter et al., 1992; Ritter et al., 1994; Dabberdt et al., 1993). Studies over the last 10-15 years have developed continuous wavelet transform (CWT) analysis to calculate spatially resolved fluxes from airborne measurements (e.g., Mauder et al., 2007; Karl et al., 2009; Metzger et al., 2013; Karl et al., 2013; Misztal et al., 2014; Yuan et al., 2015; Wolfe et al., 2015; Vaughan et al., 2017; Sayres et al., 2017; Desjardins et al., 2018; Wolfe et al., 2018; Hannun et al., 2020), including dedicated aircraft campaigns (e.g., BOREAS, CABERNET, CARAFE, OPFUE) and platforms (e.g., FOCAL). Spatial resolutions of a few km are typically achieved with good accuracy.

The final sentence of the Introduction was correspondingly extended:

[...] We explore that capability of the instrument, including the use of the CWT method, as well as the capability of the airborne eddy flux data to infer area emission rates of  $NH_3$  attributed to agriculture in rural Oklahoma. Flux uncertainties are now extensively treated in the new Sect. 3.8.

8. It is unclear if the W2018 flux toolkit was blindly used or if the investigators were aware of factors affecting systematic and random errors and if the corrections for the systematic error have been made. Given the high altitude and short legs these errors are likely rather high. Because the paper is making quantified estimates, I would strongly recommend to include the calculation of those errors as well as flux specific detection limits (e.g., as 3 x s.d. of the covariance noise far away from the lag-time).

Both systematic errors (including low- and high-frequency under-sampling) and random errors are now estimated in Sect. 3.8, including a compilation of error statistics for RF13 in Table 1. We ended up not bothering about correcting for (estimated) systematic errors of known sign, mainly because (for spatially resolved fluxes) they were comparatively small compared to random errors; e.g., often remaining of the order of ~100% even after averaging to 1 km spatial resolution. This issue is now analyzed, discussed, and visualized (new Fig. 8, modified now-Fig. 11). Reference to those uncertainties is also made in several parts of Sect. 3.9 (Flux footprint analysis, which contains the most relevant quantitative statements), and in the Summary/Conclusions.

Statements regarding the benefits of low and long legs are included as well.

9. Fig. S4, why was the lag time negative? Is it because the vertical wind data were not synchronized? What was the actual residence time?

Note that there was an error in the abscissa scales of the lag-covariance plots (top row) in Fig. S4. That mistake is now fixed, halves the mean lag times shown. We also expanded the caption to clarify that: Covariance maximized close to zero delay relative to the wind data ( $-0.01 \pm 0.74$  s); consequently, zero lag was used for this flight.

**And in Section 3.6:**

We applied lag time as obtained by the EA flux calculations (typically < 1 s; zero lag was used for RF13; see Fig. S4). Indeed; the most likely cause for *negative* lag times would be issues with synchronization. All data acquisition (DAQ) devices onboard were synchronized prior to flights. The CIMS computer clock (and hence its DAQ) synced to  $\pm 0.003$  s or better, but usually drifted by up to  $\pm 0.1$  s per hour.

The residence time in the CIMS inlet would slightly bias towards *positive* lag times. For 22 L min-1 through a 40 cm long tube with ID of 1.6 cm, the mean residence time was 0.22 s, but as a laminar flow profile would develop and the IMR core-sampled the inlet flow, the residence time of the sample would be only between 0.1 and 0.2 s. The wind measurements (AIMMS-20) were located <5 m ahead of the CIMS inlet, possibly adding <0.05 s of delay.

The caption of Fig. S4 now contains some of that too:

The likely main cause of lag was drifts in data acquisition clocks. Times were synchronized (±0.003 s or better) preflight, but net drifts of up to ±0.1 s per hour were commonly found post-flight.

10. The cospectra , Fig S4, center row, are too short to evaluate LF losses. Could both FFT and wavelet cospectra be shown for a relatively long flight leg? It would also be elegant to include in the methods how the data was filled, stitched, interpolated after removing zero air measurements.

That is true for Fig. S4 (center), because these results are for ensemble-average eddy covariance using 2-min intervals. However, both FFT and CWT co-spectra for the full legs are shown in now-Fig. S6, thus extending to almost 0.001 Hz or 90 km. NH3 measurements during the HI-SCALE campaign did not include any (substantially) longer straight-and-low-level flight legs. However, the co-spectra (Fig. S6; also Fig. 7)

suggest that LF losses, when using the CWT method, were small, as they tended to taper off towards LF. This impression is also in agreement with theory-based estimations of the systematic bias that may be induced from under-sampling LF, estimated as  $\leq 3\%$  (Sect. 3.8, Table 1).

The employed gap-filling method is described in Sect. 3.6 (3rd paragraph). The method is described in great detail in Wolfe et al. (2018), as referenced. And as described, we conservatively discarded results near the gaps. We therefore believe that the manuscript, as is, contains sufficient detail about the gap filling. We did not gain any new insights compared to what was shown by Wolfe et al. (2018), in particular their supplement.

11. I really like the research making reflection on the safety of the reagent ion. Indeed, toluene could be a much safer alternative if it works similarly well for NH3. It would be useful to add how the exhaust was routed outside of the cabin or through a VOC trap to prevent exposure.

**Added:**

To avoid exposure, the instrument exhaust was routed outside of the aircraft cabin, or into a fume hood exhaust when in the lab. The risk of spillage remained, especially when refilling the benzene reservoir in field settings.

**Technical**

• Fig 7, the color of the outside-coi CWT line cannot be easily discerned from the full-scale CWT line. Changed to black for better visibility.

• L29, L44, L371 10s can be confusing with 10 s, I suggest using "tens". Done.

**\*\*\*\*\**

**References:**

- Lee B. H., Lopez-Hilfiker F. D., Mohr C., Kurtén T., Worsnop D. R., Thornton J. A.: An Iodide-Adduct High-Resolution Time-of-Flight Chemical-Ionization Mass Spectrometer: Application to Atmospheric Inorganic and Organic Compounds, *Environ. Sci. Technol.*, 48, 6309-6317 (2014).
- Lee B. H., Lopez-Hilfiker F. D., Veres P. R., McDuffie E. E., Fibiger D. L., Sparks T. L., Ebben C. J., Green J. R., Schroder J. C., Campuzano-Jost P., Iyer S., D'Ambro E. L., Schobesberger S., Brown S. S., Wooldridge P. J., Cohen R. C., Fiddler M. N., Bililign S., Jimenez J. L., Kurtén T., Weinheimer A. J., Jaegle L., Thornton J. A.: Flight Deployment of a High-Resolution Time-of-Flight Chemical Ionization Mass Spectrometer: Observations of Reactive Halogen and Nitrogen Oxide Species, *J. Geophys. Res. Atmos.*, 123, 7670-7686 (2018).
- Wolfe G. M., Kawa S. R., Hanisco T. F., Hannun R. A., Newman P. A., Swanson A., Bailey S., Barrick J., Thornhill K. L., Diskin G., DiGangi J., Nowak J. B., Sorenson C., Bland G., Yungel J. K., Swenson C. A.: The NASA Carbon Airborne Flux Experiment (CARAFE): instrumentation and methodology, *Atmos. Meas. Tech.*, 11, 1757-1776 (2018).

**\*\*\*\*\**

New/modified figures and major text additions:

**Improved Fig. 3:**